# Hyperactivation of ERK by multiple mechanisms is toxic to RTK-RAS mutation-driven lung adenocarcinoma cells

Arun M Unni[1]*, Bryant Harbourne[2†], Min Hee Oh[2†], Sophia Wild[2], John R Ferrarone[1], William W Lockwood[2,3‡]*, Harold Varmus[1‡]*

[1]Meyer Cancer Center, Weill Cornell Medicine, New York, United States; [2]Department of Integrative Oncology, British Columbia Cancer Agency, Vancouver, Canada; [3]Department of Pathology and Laboratory Medicine, University of British Columbia, Vancouver, Canada

**Abstract** Synthetic lethality results when mutant KRAS and EGFR proteins are co-expressed in human lung adenocarcinoma (LUAD) cells, revealing the biological basis for mutual exclusivity of *KRAS* and *EGFR* mutations. We have now defined the biochemical events responsible for the toxic effects by combining pharmacological and genetic approaches and to show that signaling through extracellular signal-regulated kinases (ERK1/2) mediates the toxicity. These findings imply that tumors with mutant oncogenes in the RAS pathway must restrain the activity of ERK1/2 to avoid toxicities and enable tumor growth. A dual specificity phosphatase, DUSP6, that negatively regulates phosphorylation of (P)-ERK is up-regulated in EGFR- or KRAS-mutant LUAD, potentially protecting cells with mutations in the RAS signaling pathway, a proposal supported by experiments with *DUSP6*-specific siRNA and an inhibitory drug. Targeting DUSP6 or other negative regulators might offer a treatment strategy for certain cancers by inducing the toxic effects of RAS-mediated signaling.

DOI: https://doi.org/10.7554/eLife.33718.001

*For correspondence:
aru2001@med.cornell.edu (AMU);
wlockwood@bccrc.ca (WWL);
varmus@med.cornell.edu (HV)

†These authors contributed equally to this work
‡These authors also contributed equally to this work

**Competing interests:** The authors declare that no competing interests exist.

## Introduction

Extensive characterization of cancer genomes has begun to change the classification of neoplasms and the choice of therapies (*Garraway and Lander, 2013*). The genetic profiles of most cancers are notoriously heterogeneous, often including thousands of mutations affecting genes with a wide range of credentials—from those well-known to drive oncogenic behavior to those not known to have a role in pathogenesis. Moreover, cancers continue to accumulate mutations during carcinogenesis, producing tumor subclones with selectable features such as drug resistance or enhanced growth potential (*McGranahan and Swanton, 2017*).

Despite this heterogeneity, consistent patterns have been observed, such as the high frequency of gain-of-function or loss-of-function mutations affecting specific proto-oncogenes or tumor suppressor genes in cancers that arise in certain cell lineages. Conversely, coincident mutations in certain genes are rare, even when those genes are frequently mutated individually in specific types of cancer (*Kandoth et al., 2013*). Examples of these 'mutually exclusive' pairs of mutations have been reported in a variety of cancers (*Yoshida et al., 2011*; *Unni et al., 2015*; *Petti et al., 2006*; *Sensi et al., 2006*; *Varmus et al., 2016*); the mutual exclusivity has usually been attributed either to a loss of a selective advantage of a mutation in one gene after a change in the other has occurred ('functional redundancy') or to the toxicity (including 'synthetic lethality') conferred by the coexistence of both mutations in the same cells.

We recently reported that the mutual exclusivity of gain-of-function mutations of *EGFR* and *KRAS*, two proto-oncogenes often individually mutated in lung adenocarcinomas (LUADs), can be explained by such synthetic toxicity, despite the fact that products of these two genes operate in overlapping signaling pathways and might have been mutually exclusive because of functional redundancies (*Unni et al., 2015*). Support for the idea that the mutual exclusivity of *KRAS* and *EGFR* mutations is synthetically toxic in LUAD cells was based largely on experiments in which we used doxycycline (dox) to induce expression of mutant *EGFR* or *KRAS* alleles controlled by a tetracycline (tet)-responsive regulatory apparatus in LUAD cell lines containing endogenous mutations in the other gene (*Unni et al., 2015*). When we forced mutual expression of the pair of mutant proteins, the cells exhibited signs of RAS-induced toxicity, such as macropinocytosis and cell death. In addition, we observed increased phosphorylation of several proteins known to operate in the extensive signaling network downstream of RAS, implying that excessive signaling, driven by the conjunction of hyperactive EGFR and KRAS proteins, might be responsible for the observed toxicity.

Recognizing that such synthetic toxicities might be exploited for therapeutic purposes, we have extended our studies of signaling via the EGFR-RAS axis, with the goal of better understanding the biochemical events that are responsible for the previously observed toxicity in LUAD cell lines. In the work reported here, we have used a variety of genetic and pharmacological approaches to seek evidence that identifies critical mediators of the previously observed toxicities. Based on several concordant findings, we argue that activation of extracellular signal-regulated kinases (ERK1 and ERK2), serine/threonine kinases in the EGFR-RAS-RAF-MEK-ERK pathway, is a critical event in the generation of toxicity, and we show that at least one feedback inhibitor of the pathway, the dual specificity phosphatase, DUSP6, is a potential target for therapeutic inhibitors that could mimic the synthetic toxicity that we previously reported.

## Results

### Synthetic lethality induced by co-expression of mutant KRAS and EGFR is mediated through increased ERK signaling

In previous work, we established that mutant EGFR and mutant KRAS are not tolerated in the same cell (synthetic lethality), by placing one of these two oncogenes under the control of an inducible promoter in cell lines carrying a mutant allele of the other oncogene. These experiments provided a likely explanation for the pattern of mutual exclusivity in LUAD (*Unni et al., 2015*). While we documented several changes in cellular signaling upon induction of the second oncogene to produce toxicity, we did not establish if there is a node (or nodes) in the signaling network sensed by the cell as intolerable when both oncoproteins are produced. If such a node exists, we might be able to prevent toxicity by down-modulating the levels of activity; conversely, we might be able to exploit identification of that node to compromise or kill cancer cells.

To seek critical nodes in the RAS signaling pathway, we extended our previous study using the LUAD cell line we previously characterized (PC9, bearing the EGFR mutation, E746_A750del) and two additional LUAD lines, H358 and H1975. H358 cells express mutant KRAS (G12C), and H1975 cells express mutant EGFR (L858R/T790M). As in our earlier work, we introduced tet-regulated, mutant *KRAS* (G12V) into these lines to regulate mutant KRAS in an inducible manner and used the same vector encoding GFP rather than KRAS as a control. This single-vector system includes rtTA constitutively expressed from a ubiquitin promoter, allowing us to induce KRAS with the addition of dox (*Meerbrey et al., 2011*).

KRAS or GFP were appropriately induced after adding dox to the growth medium used for these cell lines (*Figure 1A*). To establish whether induction of a mutant *KRAS* transgene is detrimental to H358 cells producing endogenous mutant KRAS or H1975 cells producing mutant EGFR proteins, we cultured cell lines in dox for 7 days and measured the relative numbers of viable cells with Alamar blue. As we previously showed, the number of viable PC9 cells is reduced by inducing mutant KRAS (*Figure 1A*). Similarly, when mutant KRAS was induced in either H358 or H1975 cells for seven days, we observed fewer viable cells compared to cells grown without dox or to cells in which GFP was induced (*Figure 1A*). These results indicate that increased activity of the RAS pathway, either in LUAD cells with an endogenous *KRAS* mutation (H358 cells) or with an endogenous *EGFR* mutation (PC9 and H1975 cells) is toxic to these cell lines.

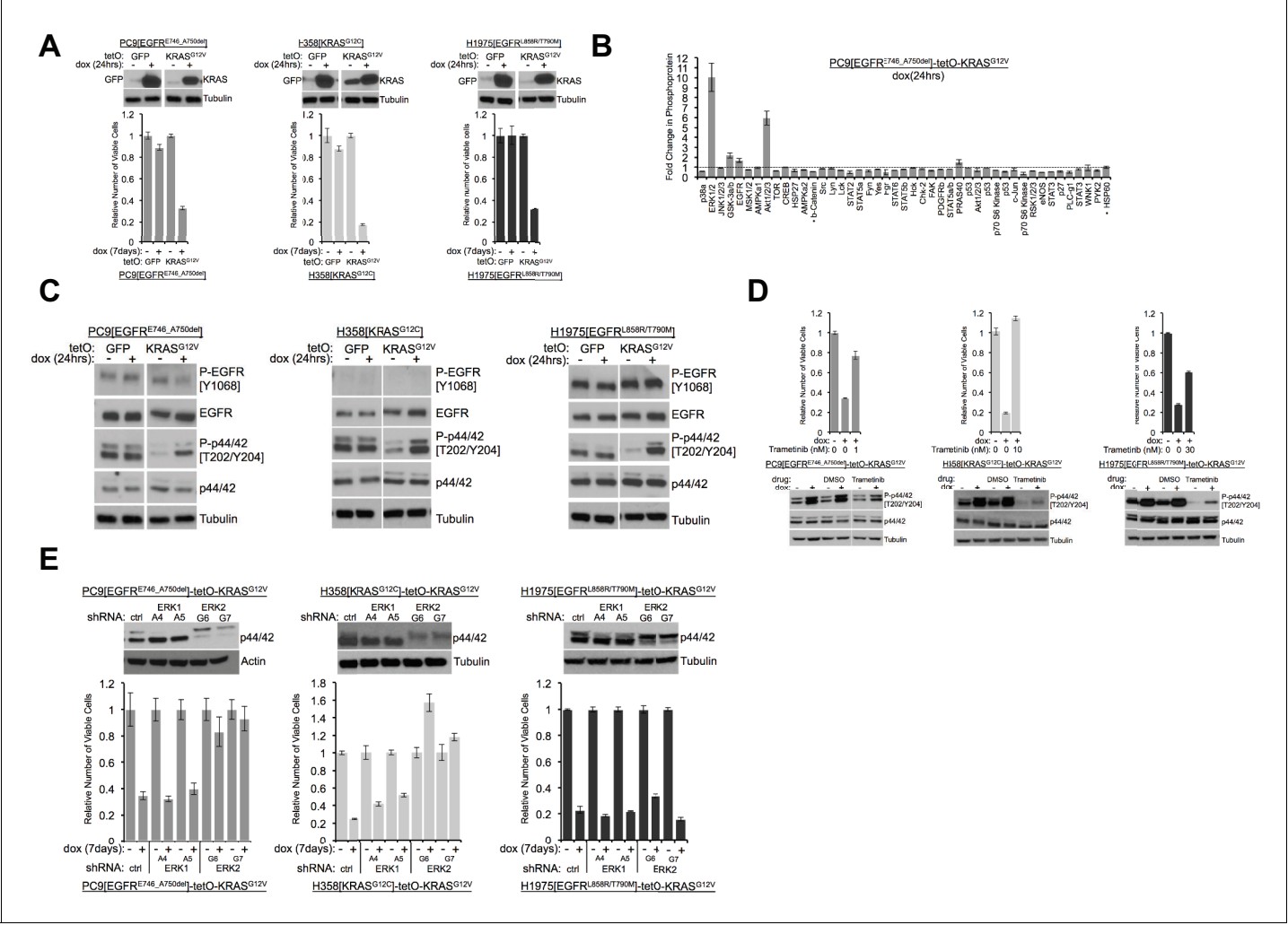

**Figure 1.** Induction of mutant KRAS reduces the numbers of viable lung cancer cells harboring KRAS or EGFR mutations, and the effects can be rescued by inhibiting ERK (A) Reduced numbers of viable LUAD cells after activation of KRAS. Production of GFP or KRAS$^{G12V}$ was induced by addition of 100 ng/mL dox in the indicated three cell lines as described in Methods. GFP and KRAS protein levels were measured by Western blotting 24 hr later. (top); tubulin served as a loading control. The numbers of viable cells, normalized to cells grown in the absence of dox (set to 1.0), were determined by measuring with Alamar blue six days later. Error bars represent standard deviations based on three replicates. (B) Induction of KRAS$^{G12V}$ uniquely increases phosphorylation of ERK1/2 among several phosphoproteins. PC9-tetO-KRAS cells were treated with dox for 24 hr and cell lysates incubated on an array to detect phosphorylated proteins. Fold changes of phosphorylation compared with lysates from untreated cells (set to 1.0, dotted line) to treated cells is presented from a single antibody array. Error bars are derived from duplicate spots on antibody array. The detection of HSP60 and ß-catenin are of total protein, not phosphoprotein. (C) Phosphorylation of ERK occurs early after induction of mutant KRAS. Lysates prepared as described for panel (A) were probed for the indicated proteins by western blot. Loading control is the same as in A. (D) Drug-mediated inhibition of the MEK1/2 kinases ameliorates KRAS-induced loss of viable cells. Mutant KRAS was induced with dox in the three indicated cell lines in the absence and presence of trametinib at the indicated dose for 7 days. The relative number of viable cells was measured with Alamar blue. Error bars represent standard deviations determined from three samples grown under each set of conditions. Values are normalized to measurements of cells that received neither dox nor trametinib (bottom). Cells were treated with dox and with or without trametinib for 24 hr at the dose conferring rescue of numbers of viable cells. Lysates were probed for indicated proteins to confirm inhibition of MEK. (E) Reduction of ERK proteins with inhibitory small hairpin (sh) RNAs protects cells from loss of viability in response to induction of mutant KRAS. LUAD cell lines, transduced with the indicated shRNA targeted against ERK1 or ERK2, were assessed for levels of ERK proteins, p42 and p44, by Western blotting (top panels). The same lines were treated with dox for 7 days and the number of viable cells measured with Alamar blue. Values are normalized to numbers of viable cells of each type grown in the absence of dox (1.0), with error bars representing standard deviations among three replicates. Similar results were obtained from 2 or 3 independent experiments.

DOI: https://doi.org/10.7554/eLife.33718.002

The following figure supplement is available for figure 1:

**Figure supplement 1.** Letality induced by mutant KRAS induction is rescued by supression of ERK.

*Figure 1 continued on next page*

*Figure 1 continued*

DOI: https://doi.org/10.7554/eLife.33718.003

We previously documented increases in phosphorylated forms of the stress kinases, phospho-JNK (P-JNK) and phospho-p38 (P-p38), as well as in phospho-ERK (P-ERK or P-p44/42), in one of these cell lines (PC9) 72 hr after treatment with dox (*Unni et al., 2015*; *Varmus et al., 2016*). We used a phospho-protein array to assess the status of protein activation more broadly after KRAS induction, using PC9-tetO-KRAS cells after 1 and 5 days of dox treatment (*Figure 1B*, *Figure 1—figure supplement 1A*). After 5 days, we again observed increases in P-JNK, P-p38, and P-ERK (*Figure 1—figure supplement 1A*), suggesting that three major branches of the MAPK pathway are activated after extended induction of mutant KRAS. In addition, several other proteins show enhanced phosphorylation at this time. At 24 hr after addition of dox, however, only P-ERK and P-AKT show a pronounced increase (*Figure 1B*). Specifically, the stress kinases, JNK and p38, were not detected as phosphorylated proteins with the protein array. A possible interpretation of these findings is that ERK may be phosphorylated relatively soon after induction of mutant KRAS, with subsequent phosphorylation (and activation) of stress kinases and several other proteins. We also observed increased phosphorylation of ERK 24 hr after induction of mutant KRAS by western blot in all three LUAD cell lines (*Figure 1C*). In H358 and in H1975-based cell systems we observed persistently increased levels of P-ERK and, ultimately, the presence of cleaved PARP (*Figure 1—figure supplement 1B*). We previously reported multiple mechanisms of RAS-induced toxicity in PC9-tetO-KRAS cells (*Unni et al., 2015*). Based on the cleavage of PARP in the studies shown here, apoptosis appears to be at least one of the mechanisms of reduced viability in H358 and H1975 cell lines.

The results shown in *Figure 1* suggest that ERK itself could be the signaling node that causes a loss of viable cells when inappropriately activated. As one test of this hypothesis, we used trametinib (*Gilmartin et al., 2011*), an inhibitor of MEK, the kinase that phosphorylates ERK, to ask whether reduced levels of P-ERK would protect cells from the toxicity caused by induction of mutant KRAS. In all three LUAD cell lines, trametinib completely or partially rescued the loss of viable cells caused by induction of mutant KRAS by dox (*Figure 1D*, *Figure 1—figure supplement 1C*). We confirmed that doses of trametinib that protected cells from the toxic effects of seven days of treatment with dox were associated with reduced levels of P-ERK after 24 hr of induction of mutant KRAS (*Figure 1D*). A PI3K inhibitor, buparlisib, did not rescue mutant KRAS-induced lethality in H358-tetO-KRAS cells (*Figure 1—figure supplement 1D*), implying that the toxic effects of KRAS are not mediated by enhanced signaling via PI3K.

To extend these findings and further challenge the hypothesis that P-ERK is an important node in the cell signaling network downstream of KRAS that confers cell toxicity, we transduced LUAD cell lines with retroviral vectors encoding shRNAs that 'knock down' expression of ERK1 or ERK2. Using two different shRNAs for each gene, as well as a non-targeted shRNA vector as control, we stably reduced the levels of ERK1 or ERK2 in the three LUAD cell lines (*Figure 1E*). When PC9 and H358 lines were treated with dox to assess the effects of ERK1 or ERK2 knockdowns on the loss of viable cells, we found that depletion of ERK2, but not ERK1, rescued cells from KRAS toxicity after 7 days in dox (*Figure 1E*). In H1975 cells, however, neither knockdown of ERK1 nor of ERK2 prevented KRAS-induced cell toxicity. Since trametinib rescues the number of viable cells after induction of KRAS in H1975 cells (*Figure 1D*), it seemed possible that either ERK1 or ERK2 might be sufficient to mediate RAS-induced toxicity in this line. In that case, it would be necessary to reduce the levels or the activity of both ERK proteins to rescue H1975 cells from toxicity. We tested this idea by treating dox-induced H1975-tetO-KRAS cells with SCH772984 (*Morris et al., 2013*), a drug that inhibits the kinase activity of both ERK1 and ERK2 (*Figure 1—figure supplement 1E*). As we observed with the MEK inhibitor, trametinib, in other lines (*Figure 1D*, far right), the ERK inhibitor reduces KRAS-associated toxicity in H1975 cells with concomitant reductions of P-ERK1 and P-ERK2 (*Figure 1—figure supplement 1E*).

To examine this issue in a different way, we performed a genome-wide CRISPR-Cas9 screen to evaluate mechanisms of mutant KRAS-induced toxicity in an unbiased manner. After growing H358-tetO-KRAS cells for 7 days following introduction of the appropriate vectors carrying Cas9 and a library of DNA encoding gene-targeted RNAs (see Materials and methods), guide RNA (sgRNA)

targeting ERK2 (MAPK1) was highly enriched in cells grown in the presence of doxycycline (*Figure 1—figure supplement 1F*, *Supplementary file 1*). Guide RNA targeting RAF1 (CRAF) was also significantly enriched. Data from this CRISPR-Cas9 genome-wide screen strongly suggests that depletion of critical proteins in the RTK-RAS pathway can mitigate the toxicity induced by excess RAS activation. Collectively, our data suggest that LUAD cell lines are sensitive to inappropriate hyperactivation of the ERK signaling node and that toxicity mediated by activation of the RAS pathway is ERK-dependent.

## DUSP6 is a major regulator of negative feedback, expressed in LUAD cells, and associated with KRAS and EGFR mutations and with high P-ERK levels

The evidence that hyperactive ERK signaling has toxic effects on LUAD cells raises the possibility that cancers driven by mutations in the RAS pathway may have a mechanism to 'buffer' P-ERK levels and thereby avoid reaching a lethal signaling threshold. Genes encoding negative feedback regulators are typically activated at the transcriptional level by the EGFR-KRAS-ERK pathway to place a restraint on signaling (*Avraham and Yarden, 2011*). Such feedback regulators previously implicated in the control of EGFR-KRAS-ERK signaling include the six dual specificity phosphatases (DUSP1-6), the four sprouty proteins (SPRY1-4) and the three sprouty-related, EVH1 domain-containing proteins (SPRED1-3) (*Avraham and Yarden, 2011*; *Lake et al., 2016*). To begin a search for possible negative regulators of RAS-mediated signaling in LUAD cells driven by mutations in either *KRAS* or *EGFR*, we asked whether mutations in either proto-oncogene would up-regulate one or multiple members of these families of regulators, based on the assumption that such proteins might constrain P-ERK levels, leading to optimal growth without cytotoxic effects.

To search for potential negative regulators specifically involved in LUAD, we compared amounts of RNAs from *DUSP*, *SPRY* and *SPRED* gene families in tumors with and without mutations in either *KRAS* or *EGFR*, using RNA-seq data from The Cancer Genome Atlas (TCGA) (*Cancer Genome Atlas Research Network, 2014*) (*Figure 2A,B* and *Figure 2—figure supplement 1A,B*). *DUSP6* was the only negative-feedback regulatory gene with significantly different levels of expression when we compared tumors with mutations in either *KRAS* or *EGFR* with tumors without such mutations (Bonferoni corrected $p < 0.01$, two-tailed t-test with Welch's correction). Further, *DUSP6* mRNA was significantly up-regulated in LUAD tumors with mutations in common RTK-RAS pathway components compared to those without, consistent with a role of DUSP6 in regulating EGFR-KRAS-ERK signaling (*Figure 2—figure supplement 1C*) (*Avraham and Yarden, 2011*; *Muda et al., 1996a*; *Muda et al., 1996b*; *Groom et al., 1996*; *Kidger and Keyse, 2016*; *Zhang et al., 2010*). *DUSP6* RNA was also present at higher levels in LUADs with *EGFR* or *KRAS* mutations than in tumors without such mutations in an independent collection of 83 tumors collected at the British Columbia Cancer Agency (BCCA, $p = 0.004$), confirming the findings derived from the TCGA dataset (*Figure 2C* and *Figure 2—figure supplement 1D*). Furthermore, *DUSP6* RNA was more abundant in EGFR/KRAS mutant LUADs than in normal lung tissue ($p<0.0001$) whereas no significant differences in *DUSP6* levels were observed between normal lung tissue and tumors without mutations in either of these two genes ($p = 0.64$) (*Figure 2C* and *Figure 2—figure supplement 1D*).

To ascertain whether *DUSP6* is up-regulated specifically in tumors driven by mutant KRAS or mutant EGFR signaling rather than in tumors associated with activation of other oncogenic pathways, we measured *DUSP6* RNA in experimental systems driven by the activation of various oncogenes. In transgenic mouse models of lung cancer, *Dusp6* RNA was present at significantly higher levels in the lungs of mice bearing tumors driven by mutant *EGFR* or *KRAS* transgenes than in normal mouse lung epithelium (*Figure 2D*) (*Felsher and Bishop, 1999*; *Fisher et al., 2001*; *Politi et al., 2006*). In contrast, *Dusp6* RNA levels were not significantly different in lungs from mice with tumors driven by MYC and in normal mouse lung tissue (*Figure 2D*). Similarly, increased levels of *DUSP6* RNA were observed in primary human epithelial cells only when the cells were also transduced with mutant *RAS* genes, but not with a variety of other oncogenes or with plasmids encoding GFP ($p < 0.0001$) (*Figure 2E*) (*Bild et al., 2006*). Lastly, our LUAD cell lines engineered to produce KRAS$^{G12V}$ in response to dox showed an increase in *DUSP6* RNA that correlated with augmented phosphorylation of ERK and cell toxicity (*Figure 2F*). It is unclear why increased levels of *DUSP6* RNA are not sufficient to decrease P-ERK in these inducible systems; this may reflect the localization of P-ERK, which we have not explored here. Together, these findings suggest that DUSP6 is a critical

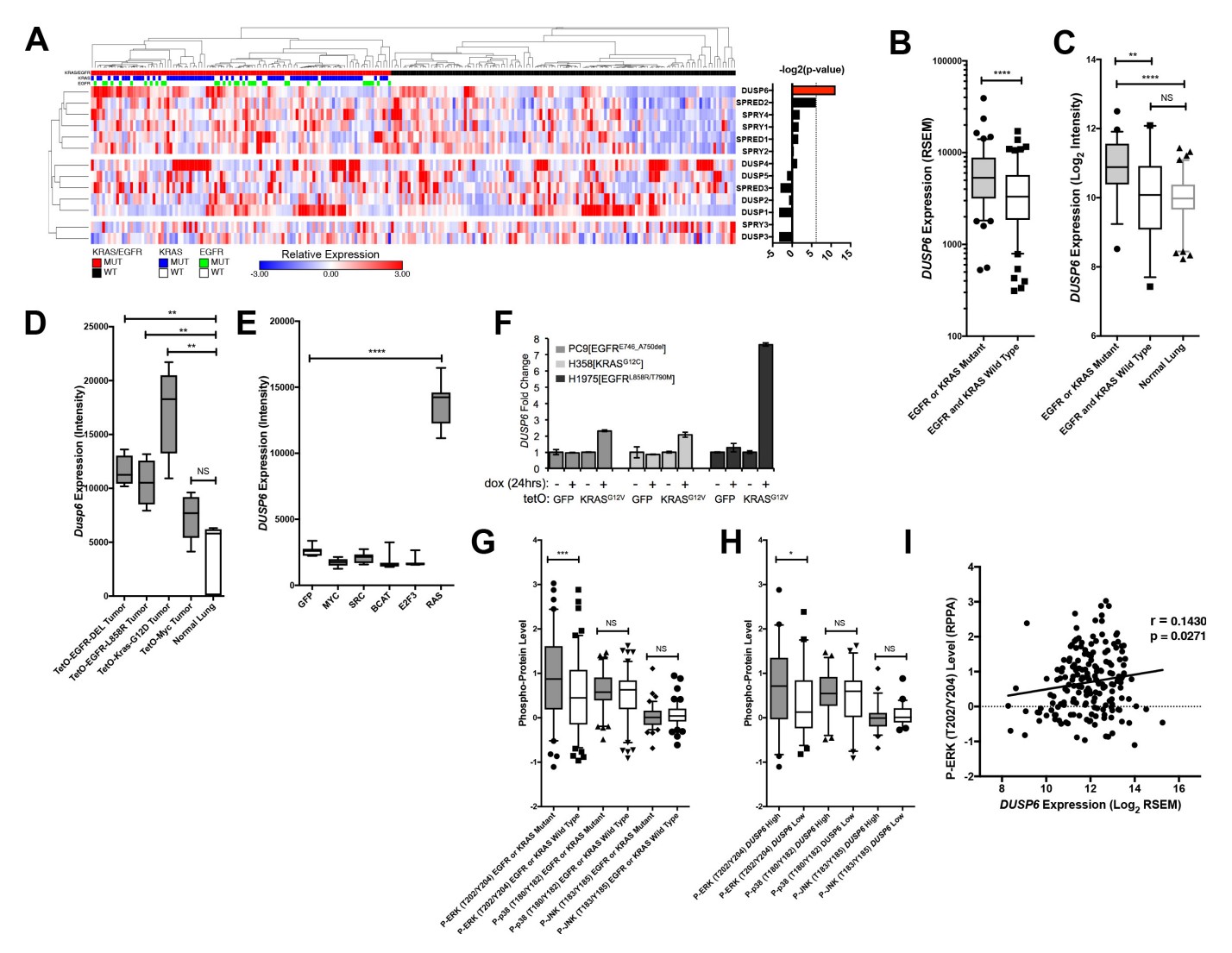

**Figure 2.** *DUSP6* is the only negative feedback regulator significantly up-regulated in LUAD tumors with KRAS or EGFR mutations. (A) Negative feedback regulators differentially expressed between clinical LUADs with or without *EGFR* or *KRAS* mutations (as indicated in green or blue, respectively, in the third and second horizontal bars). Expression levels for the indicated genes as determined by RNA-seq were compared between LUAD tumors with (n = 107, red) and without (n = 123, black) *KRAS* or *EGFR* mutations. In the heatmap, red indicates high relative expression and blue, low expression. Significance, as determined by two-tailed unpaired t-test with Bonferroni multiple testing correction, is indicated as the –log2(p-value). The significance threshold was set at a p-value < 0.01 and is indicated by the dotted line. Only *DUSP6* surpassed this threshold. (B) *DUSP6* is the main negative feedback regulator upregulated in LUADs with *EGFR* or *KRAS* mutations. Box plots show levels of *DUSP6* RNA from samples in A. LUADs with *EGFR* or *KRAS* mutations (n = 107) express *DUSP6* at higher levels than do LUADs with wildtype *KRAS* and *EGFR* (n = 123) in the TCGA dataset. (C) Validation of increased *DUSP6* expression in LUADs with mutated *KRAS* or *EGFR*. In an independent internal dataset from the BCCA, LUADs with *EGFR* or *KRAS* mutations (n = 54) demonstrated higher expression of *DUSP6* compared to LUADs in which both *EGFR* and *KRAS* were wild-type (n = 29) and to normal lung tissues (n = 83). (D) *Dusp6* is upregulated in the lungs of mice with tumors induced by mutant *EGFR* or *Kras* transgenes. Tumor-bearing lung tissues from mice expressing *EGFR* or *Kras* oncogenes produce higher levels of Dusp6 RNA than do normal lung controls or tumor-bearing lungs from mice with a *MYC* transgene. (E) Increased DUSP6 RNA is specific to cells with oncogenic signaling through RAS. Human primary epithelial cells expressing a *HRAS* oncogene (n = 10 biological replicates) express *DUSP6* at higher levels than control cells producing GFP (n = 10 biological replicates) whereas cells expressing known oncogenes other than *RAS* genes (*MYC, SRC, B-Catenin,* and *E2F-3*) do not. (F) DUSP6 RNA levels increase in PC9, H358 and H1975 cells expressing mutant KRAS. Dox was added to induce either *GFP* or the *KRAS*G12V oncogene for 24 hr; DUSP6 RNA was measured by qPCR. (G–I) DUSP6 expression is associated with P-ERK levels. (G) LUADs with *EGFR* or *KRAS* mutations (n = 107) have higher P-ERK levels, but not P-p38 or P-JNK levels, than LUADs with wildtype *KRAS* and *EGFR* (n = 123) in the TCGA dataset. (H) LUADs with the highest *DUSP6* RNA levels (n = 46) demonstrated higher P-ERK levels, but not P-p38 or P-JNK levels, than LUADs with the lowest *DUSP6* RNA levels (n = 46). (I) *DUSP6* RNA levels correlate with the levels of P-ERK in LUADs (n = 182). Pearson correlation coefficient (r) and p-value are indicated. *p < 0.05, **p < 0.01, ***p < 0.001, ****p < 0.0001, NS = Not Significant.

*Figure 2 continued on next page*

*Figure 2 continued*

DOI: https://doi.org/10.7554/eLife.33718.004

The following figure supplement is available for figure 2:

**Figure supplement 1.** Negative feedback regulators are differentially expressed in clinical LUADs with or without *EGFR* or *KRAS* mutations.

DOI: https://doi.org/10.7554/eLife.33718.005

negative feedback regulator activated in response to oncogenic signaling by mutant RAS or EGFR proteins in LUAD.

In our previous study (*Unni et al., 2015*) (see also *Figure 1—figure supplement 1A*), we found that co-induction of oncogenic KRAS and EGFR activated not only ERK, but also JNK and p38 MAPK pathways, albeit at later times. To investigate whether *DUSP6* is up-regulated solely in response to phosphorylation of ERK or also in response to phosphorylation of JNK and p38, we assessed the relationship of amounts of *DUSP6* RNA in tumors with levels of P-ERK, P-JNK and P-p38 proteins as determined for TCGA (*Cancer Genome Atlas Research Network, 2014*), using the Reverse Phase Protein Array (RPPA). LUADs with a *KRAS* or an *EGFR* mutation contained significantly higher levels of P-ERK – but not of P-JNK or P-p38 – than did tumors without those mutations, consistent with a role for these oncogenes in ERK activation (*Figure 2G*). Furthermore, tumors with high *DUSP6* RNA have relatively high amounts of P-ERK but not of P-JNK or P-p38 (*Figure 2H*). Lastly, there is a positive correlation between P-ERK levels and *DUSP6* RNA in LUAD (*Figure 2I*), whereas no such association was observed between *DUSP6* RNA and P-JNK or P-p38 (*Figure 2—figure supplement 1E,F*). Together, these observations support the proposal that *DUSP6* is expressed in response to activation of ERK and that it serves as a major negative feedback regulator of ERK signaling in LUAD, buffering the potentially toxic effects of ERK hyperactivation.

## Knockdown of DUSP6 elevates P-ERK and reduces viability of LUAD cells with either KRAS or EGFR oncogenic mutations

If DUSP6 is a negative feedback regulator of RAS signaling through ERK, then inhibiting the function of DUSP6 in LUAD cell lines driven by oncogenic KRAS or EGFR should cause hyperphosphorylation and hyperactivity of ERK, possibly producing a signaling intensity that causes cell toxicity, as observed when we co-express mutant KRAS and EGFR. Consistent with this prediction, introduction of *DUSP6*-specific siRNA pools into PC9 cells decreased DUSP6 levels and reduced the number of viable cells to levels similar to those observed when mutant *EGFR*, the driver oncogene, was itself knocked down (*Figure 3A*). siRNA pools for either *DUSP6* or *EGFR* decreased DUSP6 protein levels. A decrease in DUSP6 protein levels with siRNA against *EGFR* RNA can be explained by a reduction in EGFR protein levels causing a decrease in ERK activation (*Figure 3A*) and subsequently diminishing expression of *DUSP6*, a direct negative feedback regulator of ERK activity. Importantly, almost complete knockdown of DUSP6 was required to elicit toxic effects in PC9 cells.

The pool of Dharmacon-synthesized siRNAs we used is composed of 4 individual siRNAs (labeled DUSP6-6,7,8 and 9, *Figure 3* and *Figure 3—figure supplement 1A,B*). We tested the individual siRNAs to confirm knockdown of DUSP6 protein and assess cell viability after siRNA treatment (*Figure 3—figure supplement 1A,B*). Treatment of PC9 cells with any one of three particular siRNAs resulted in a significant decrease in DUSP6 levels (particularly DUSP6-6 and DUSP6-7), however, the number of viable cells on day 5 was greater than in cells treated with the non-targeting control siRNA (*Figure 3—figure supplement 1A,B*). This observation was in contrast to the loss of cell viability we documented with the siRNA pool against DUSP6 (*Figure 3*). However, treatment with one other siRNA in the pool, DUSP6-8, resulted in the greatest depletion in DUSP6 protein and also a striking loss of cell viability (*Figure 3—figure supplement 1A,B*), consistent with the results from the siRNA pool. This suggests that DUSP6 protein levels need to be substantially depleted to exert an effect in PC9 cells.

Because only one siRNA in the pool (DUSP6-8) had a deleterious effect on PC9 cells, we confirmed the effects of this siRNA by utilizing another siRNA that targets a different region of DUSP6 mRNA (A 5' coding sequence is targeted by DUSP6-Qiagen, whereas a 3' coding sequence is targeted by DUSP6-8). DUSP6-Qiagen suppresses DUSP6 protein to a level similar to what we observed with the siRNA pool (*Figure 3B,C*). We also observed a loss of cell viability in PC9s cells

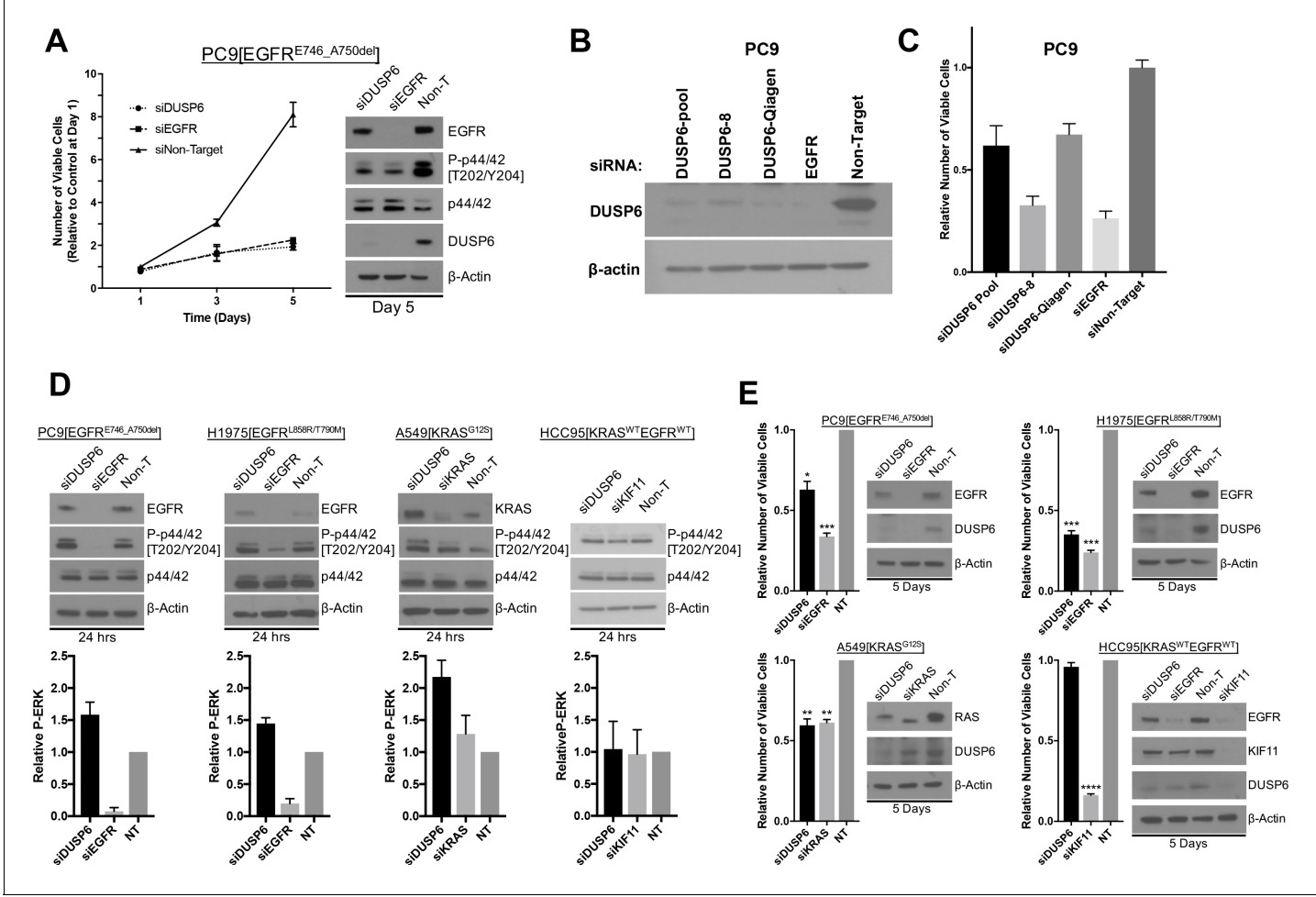

**Figure 3.** Knockdown of DUSP6 increases P-ERK and selectively inhibits LUAD cell lines with KRAS or EGFR mutations. (**A**) Interference with *DUSP6* RNA induces toxicity in PC9 cells. Pooled siRNAs for *DUSP6, EGFR* or a non-gene targeting control (Non-T) were transfected into PC9 cells (carrying an *EGFR* mutation) on day 0 and day 3, and the numbers of viable cells in each condition was measured with Alamar blue at the indicated time points and scaled to the Non-T condition at day 1 to measure the relative changes in numbers of viable cells. Experiments were done in biological triplicate with the average values presented ±SEM. Western blots were performed at the endpoint of the assay (day 5) to confirm reduced amounts of DUSP6 protein and measure levels of ERK and P-ERK (p42/44 and P-p42/44, respectively). (**B–C**) A siRNA that targeted the 5' region of DUSP6 mRNA coding sequence (siDUSP6-Qiagen; different from siDUSP6-8 that targets the 3' mRNA coding region), reduces levels of DUSP6 protein and decreases the numbers of viable cells. The indicated siRNAs (DUSP6-pool, DUSP6-8, DUSP6-Qiagen, EGFR and Non-Target) were delivered to PC9 cells, the levels of DUSP6 protein measured and the numbers of viable cells was determined as described for panel A. Experiments were done at least three times, and the average ±SEM is indicated for cell viability. (**D**) Interference with *DUSP6* RNA acutely increases P-ERK levels. DUSP6 was knocked down in PC9 and H1975 cells (*EGFR* mutants), A549 cells (*KRAS* mutant), and HCC95 cells (*KRAS* and *EGFR* wild-type); levels of ERK and P-ERK were measured by Western blot 24 hr later. Relative P-ERK levels (ratio of phosphorylated to total levels normalized to actin) were determined by dosimetry and compared to the non-targeting control (NT) to quantify the relative increase after DUSP6 knockdown. Three independent western blots were performed and the average ±SEM is plotted. (**E**) Interference with *DUSP6* RNA inhibits LUAD cell lines with activating mutations in genes encoding components of the EGFR/KRAS signaling pathway. Numbers of viable cells 5 days after knockdown of DUSP6 or knockdown of positive controls (EGFR, KRAS or KIF11) were assessed with Alamar blue and compared to the non-targeting controls to determine relative changes. Experiments were done in biological triplicate with the average values presented ±SEM. Western blots to monitor knockdown of target genes at Day 5 are also displayed. *p < 0.05, **p < 0.01, ***p < 0.001, ****p < 0.0001, NS = Not Significant.

DOI: https://doi.org/10.7554/eLife.33718.006

The following figure supplement is available for figure 3:

**Figure supplement 1.** (A–B) Extensive knockdown of expression of *DUSP6* with individual or pooled siRNAs is necessary to induce toxicity in PC9 cells.
DOI: https://doi.org/10.7554/eLife.33718.007

treated with DUSP6-Qiagen siRNA comparable to that of the siRNA pool, suggesting these effects are not off-target (*Figure 3B,C*).

While it was anticipated that knockdown of mutant EGFR would diminish the numbers of viable cells by reducing levels of P-ERK and its growth-promoting signal, cells in which DUSP6 was knocked down with siRNAs also displayed reduced P-ERK levels five days after transfection, not the expected increase in phosphorylation of ERK (*Figure 3A*). One way to reconcile this apparent discrepancy is to examine the kinetics of phosphorylation and dephosphorylation of ERK after manipulation of the abundance of DUSP6 and its resulting effects on RAS signaling. To determine whether an initial, transient increase in P-ERK occurred after nearly complete knockdown of DUSP6, preceding the observed reduction in viable cells, we measured P-ERK in two cell lines with mutations in *EGFR* (PC9 and H1975 cells), one cell line with a mutation in *KRAS* (A549 cells) and a lung squamous cell carcinoma with wildtype *EGFR* and *KRAS* (HCC95 cells) 24 hr after addition of DUSP6 siRNA. In the three cell lines assessed with mutant *EGFR* or *KRAS*, there was a small but consistent increase (~1.5 fold) in P-ERK 24 hr after receiving DUSP6 siRNA, compared to non-targeting siRNA controls (*Figure 3D*). Within 5 days, knockdown of DUSP6 reduced the numbers of viable cells in the LUAD lines with activating *KRAS* or *EGFR* mutations (PC9, H1975 and A549 cells), but not in a cell line with no known activating mutations affecting the EGFR-KRAS-ERK pathway (HCC95 cells) (*Figure 3E*).

Mirroring the decrease in viability, cleaved PARP was also induced five days after DUSP6 knockdown in EGFR/KRAS mutant, but not EGFR/KRAS wildtype cells (*Figure 3—figure supplement 1C*). While there was no correlation between sensitivity to DUSP6 knockdown and basal DUSP6 protein levels, KRAS or EGFR mutant cell lines demonstrate higher P-ERK levels and/or a high P-ERK to DUSP6 protein ratio that could contribute to P-ERK hyperactivity and the subsequent decrease in cell viability after inhibition of DUSP6 (*Figure 3*-figure supplement D,E,F). Lastly, as described above, reduction of ERK1 or ERK2 levels with shRNAs in EGFR-mutant PC9 cells partially rescued the decreased cell viability caused by DUSP6 knockdown, suggesting that ERK – at least in part - mediates the toxic effects of DUSP6 inhibition (*Figure 3—figure supplement 1G,H,I*). These data suggest that knockdown of DUSP6 or potentially other negative feedback regulators that can increase P-ERK would reduce cell viability in cells containing an oncogenic *KRAS* or *EGFR* mutation.

## Pharmacological inhibition of DUSP6 reduces the number of viable LUAD cells bearing mutations that activate the ERK pathway

The results presented thus far suggest that LUAD cells with mutations in *KRAS* or *EGFR* depend on negative regulators like DUSP6 to attenuate P-ERK for survival, offering a potentially exploitable vulnerability that could be useful therapeutically. However, blocking synthesis of DUSP6 efficiently with siRNA is difficult, in part because reduced levels of DUSP6 lead to increased levels of phosphorylated ERK, stimulating a subsequent increase in *DUSP6* mRNA. As *DUSP6* mRNA rises, more siRNA may be required to sustain the reduction of DUSP6. Based on this negative feedback cycle, we reasoned that pharmacological inhibition of the enzymatic activity of DUSP6 would be more effective. A small molecule inhibitor of DUSP6, (E)−2-benzylidene-3-(cyclohexylamino)−2,3-dihydro-1H-inden-1-one (BCI), was identified through an in vivo chemical screen for activators of fibroblast growth factor signaling in zebrafish (*Molina et al., 2009*; *Korotchenko et al., 2014*). BCI inhibits DUSP6 allosterically, binding near the active site of the phosphatase, inhibiting activation of the catalytic site after binding to its substrate, ERK (*Molina et al., 2009*). BCI also selectively inhibits DUSP1, which, like DUSP6, has catalytic activity dependent on substrate binding. However, as demonstrated in *Figure 2A*, *DUSP1* is not significantly up-regulated in LUADs with *EGFR* or *KRAS* mutations. Furthermore, siRNA-mediated knockdown of DUSP1, as opposed to knockdown of DUSP6, has no effect on viability of EGFR-mutant H1975 cells, suggesting that DUSP6 should be the main target of BCI (*Figure 4—figure supplement 1A,B*).

We tested 11 lung cancer cell lines - 8 with a *KRAS* or *EGFR* mutation and 3 with no known activating mutations in these genes – with a dosing strategy covering the previously determined active range of the drug (*Shojaee et al., 2015*). We predicted that cancer lines with mutations in *KRAS* or *EGFR* would be more sensitive to the potential effects of BCI treatment on numbers of viable cells, since DUSP6 would be required to restrain the toxic effects of P-ERK in these cells. Our findings are consistent with this prediction (*Figure 4A,B*). The cell lines fell into three categories of sensitivity: 1) the most sensitive lines, with IC50s between 1–3 uM and with > 90% loss of viable cells at 3.2 uM, all harbored *KRAS* or *EGFR* mutations; 2) the one line with intermediate sensitivity, H1437 (IC50 > 4

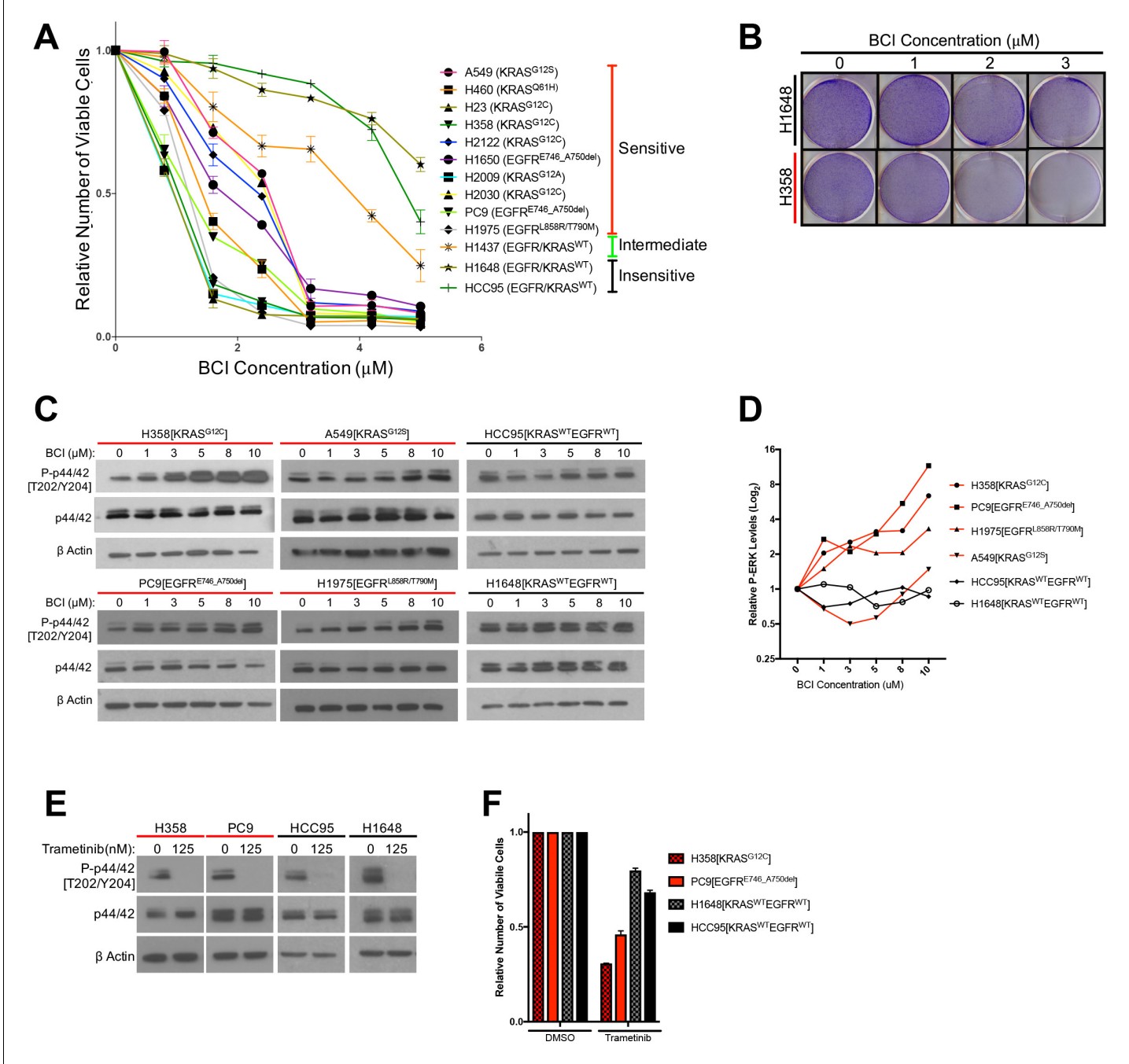

**Figure 4.** Treatment with the DUSP6 inhibitor BCI selectively kills LUAD cell lines with KRAS or EGFR mutation, implying a dependence on ERK-mediated signaling. (**A–B**) BCI induces toxicity specifically in lung cancer cell lines with mutations in genes encoding components in the EGFR-KRAS-ERK pathway. (**A**) Eleven lung cancer cell lines were treated with increasing doses of BCI for 72 hr based on the reported effective activity of the drug (*Shojaee et al., 2015*). Cell lines could be assigned to three distinct groups: sensitive (red), intermediate (green) and insensitive (black). All sensitive cell lines contained either *EGFR* or *KRAS* mutations; the intermediate and insensitive cell lines were wild-type for genes encoding components of the EGFR-KRAS-ERK signaling pathway (as determined by the Sanger Cell Line Project and the Cancer Cell Line Encyclopedia [*Barretina et al., 2012*]). Experiments were done in biological duplicate with the average values presented ±SEM. (**B**) Crystal Violet stain of cells plated in the indicated doses of BCI or control (0 = 0.1% DMSO) for 72 hr. Sensitive cells with a *KRAS* mutation (H358 cells; denoted with red underlining) show a more pronounced decrease in cell number than do cells without oncogenic mutations in genes encoding components of the EGFR-KRAS-ERK pathway (H1648 cells; black underlining). Experiments were done in biological duplicate with a representative image shown. (**C**) BCI increases P-ERK levels specifically in BCI-sensitive cell lines. Sensitive lines (H358, PC9, H1975 and A549; red underlining) and insensitive lines (HCC95 and H1648; black underlining) were treated with the indicated doses of BCI or vehicle control (0.1% DMSO) for 30 min, and the levels of ERK (p44/p42) and P-ERK (P-p44/42 T202/Y204)

*Figure 4 continued on next page*

*Figure 4 continued*
assessed by Western blot. P-ERK appeared in the sensitive cells at low doses of BCI, but P-ERK levels did not increase in the insensitive cells at the tested doses of BCI. (D) Dosimetry plots from the experiment shown in panel. (C) (E–F) Cell lines sensitive to BCI are also dependent on P-ERK for survival. BCI-sensitive cells with oncogenic mutations in *EGFR* or *KRAS* (PC9 and H358, respectively; red underlining) and BCI-insensitive cells (H1648 and HCC95; black underlining) were treated with the indicated doses of the MEK inhibitor trametinib for 72 hr; viable cells were measured with Alamar blue and compared to cells receiving the vehicle control (0 = 0.1% DMSO). (E) Treatment with trametinib decreased P-ERK levels as determined by western blot. (F) The reduction in P-ERK corresponded to a greater decrease in viable cells in BCI-sensitive lines (red coloring), compared to BCI-insensitive cell lines (black coloring).

DOI: https://doi.org/10.7554/eLife.33318.008
The following figure supplement is available for figure 4:

**Figure supplement 1.** (A–B) Knockdown of DUSP6, but not DUSP1, decreases viability of LUAD cells.
DOI: https://doi.org/10.7554/eLife.33318.009

uM), contains an activating mutation in *MEK* (Q56P); and 3) the relatively insensitive lines (IC50s $\geq$ 5 uM) lack known mutations affecting the EGFR-KRAS-ERK signaling pathway. The insensitive cell lines did not demonstrate the marked (> 90%) reduction in numbers of viable cells observed with the sensitive cell lines and only sensitive cell lines showed induction of cleaved PARP after BCI treatment (*Figure 4—figure supplement 1C*). Together, these data suggest that pharmacological inhibition of DUSP6 specifically kills cells with *EGFR* or *KRAS*-mutations.

## P-ERK levels increase in LUAD cells after inhibition of DUSP6 by BCI, and P-ERK is required for BCI- mediated toxicity

Based on findings in the preceding section, we predicted that BCI-mediated inhibition of DUSP6 would increase P-ERK to toxic levels, similar to the effects of co-expressing mutant *KRAS* and *EGFR*. To test this proposal, we measured total ERK and P-ERK after BCI treatment in sensitive and insensitive cell lines. A subset of the most sensitive cell lines, H358 (KRAS mutant) and PC9 and H1975 (EGFR mutants), demonstrated a large, dose-dependent increase in P-ERK in response to BCI treatment, with appreciable increases observed even at the lowest doses tested (1 uM) (*Figure 4C,D*). This induction of P-ERK precedes the appearance of cleaved PARP and cell death, as indicated by a time course of observations after BCI treatment in KRAS-mutant H358 cells (*Figure 4—figure supplement 1D*). Likewise, another sensitive cell line, A549 (KRAS mutant), demonstrated an increase in P-ERK, albeit at higher BCI concentrations, consistent with a less acute BCI sensitivity (*Figures 3C and 4C,D*). Conversely, BCI did not induce increases in P-ERK in the insensitive cell lines HCC95 and H1648, even at the highest levels of BCI (10 uM) (*Figure 4C,D*). Importantly, cell lines sensitive to BCI were also dependent on sustained P-ERK signaling for survival, as the MEK inhibitor trametinib, while effectively reducing P-ERK in all cell lines, reduced cell viability to a greater degree in BCI- sensitive lines (H358 and PC9) compared to BCI-insensitive lines (H1648 and HCC95; *Figure 4E,F*). Thus, the oncogenic mutation profile and dependency on activation of the EGFR-RAS-ERK pathway correlates with dependence on DUSP6 activity. These correlations are likely to reflect the central significance of P-ERK as a determinant of cell growth and viability.

To confirm whether P-ERK is involved in regulation of BCI-mediated cell death, we treated KRAS mutant H358 cells with a combination of BCI and the ERK1/2 inhibitor VX-11E, predicting that simultaneous inhibition of DUSP6 and ERK would mitigate the toxic effects of BCI treatment. Unlike other ERK inhibitors such as SCH772984, VX-11E does not block ERK phosphorylation, but instead limits ERK activity following phosphorylation (*Chaikuad et al., 2014*). Consistent with this, while no difference in P-ERK induction was observed, VX-11E treatment limited BCI- induced phosphorylation of the downstream ERK target RSK (*Figure 4—figure supplement 1F*). In addition, treatment with VX-11E lead to a relative increase in the number of viable cells after BCI treatment in a dose-dependent manner, with higher VX-11E concentrations demonstrating less decline in viability in response to BCI compared to lower doses (*Figure 4—figure supplement 1E*). Together, these data suggest that ERK activation plays a vital role in mediating the inhibitory effects of BCI treatment in KRAS or EGFR mutant lung cancer cells.

To further understand BCI-mediated toxicity, we searched for potential resistance mechanisms through an unbiased, genome-wide CRISPR screen of the type described earlier (*Figure 1—figure supplement 1F*). If loss of genes targeted by guide RNA confers resistance, that can reveal the

nature of the pathway being targeted, since inhibited expression of the gene mitigates the effects of the drug. We performed this screen in H460 cells that are mutant (Q61H) for KRAS and sensitive to BCI (*Figure 4A*). In the screen, we found that sgRNAs targeting KRAS were significantly enriched in *KRAS*-mutated H460 cells upon treatment with BCI compared to untreated controls (*Figure 4—figure supplement 1G*, *Supplementary file 1*). Guide RNA targeting *KRAS* were depleted in the absence of drug suggesting a dependence on mutant KRAS in this cell line. These results suggest that KRAS pathway activity is a major determinant of sensitivity to BCI (*Figure 4—figure supplement 1G*). To validate these results, we cloned two individual sgRNAs targeting *KRAS* and transduced H460 cells. After 7 days of puromycin selection, the polyclonal population was evaluated for KRAS depletion (*Figure 4—figure supplement 1H*). The KRAS-targeted and control H460 cells were treated at this time point with a dose response of BCI for 72 hr. Cells that contained sgRNAs against *KRAS* were less sensitive to BCI than cells containing control sgRNA and un-manipulated cells (*Figure 4—figure supplement 1I*).

We also generated two clones of DUSP6-deficient H358 cells using CRISPR-Cas9 and independent guide RNAs (*Figure 4—figure supplement 1J*). Unexpectedly, both clones remained responsive to BCI's cell killing activity (*Figure 4—figure supplement 1K*). These results may be explained by the presence of DUSP1 (*Figure 4—figure supplement 1J*) and the reported activity of BCI against DUSP1 in addition to DUSP6. Further studies will be required to ascertain if these cells are still dependent on P-ERK for BCI-mediated sensitivity through DUSP1 or through another mechanism. While BCI sensitivity may not be solely due to DUSP6, our genome-wide screen for resistance to BCI suggests activation of the RAS pathway is at least partly required.

To further test RAS pathway dependency and its relation to BCI sensitivity, we predicted that stimulating the EGFR-RAS-ERK pathway in a BCI-insensitive cell line would make the cells more dependent on DUSP6 activity and more sensitive to BCI. Using HCC95 lung squamous carcinoma cells, which express relatively high levels of wild-type EGFR (*Figure 5A*), we showed that EGF increased the levels of both P-EGFR and P-ERK, confirming activation of the relevant signaling pathway (*Figure 5A,B*, *Figure 5—figure supplement 1*). In addition, BCI further enhanced the levels of P-ERK, especially in the EGF-treated cells, with dose-dependent increases; these findings are similar to those observed in cell lines with *EGFR* or *KRAS* mutations (*Figure 4C,D*). After pretreatment with EGF (100 ng/mL) for ten days and treating the cells with increasing doses of BCI to inhibit DUSP6, 3 uM BCI reduced the number of viable HCC95 cells by approximately 40% compared to the control culture that did not receive EGF (*Figure 5C*). This outcome implies that prolonged EGF treatment and subsequent activation of P-ERK signaling makes HCC95 cells dependent on DUSP6 activity, as also observed in cell lines with *EGFR* or *KRAS* mutations (*Figure 4A*). Taken together, these findings suggest that LUAD cells with *KRAS* or *EGFR* mutations are sensitive to BCI because the drug acutely increases P-ERK beyond a tolerable threshold in a manner analogous to the synthetic lethality we previously described in LUAD lines after co-expression of mutant KRAS and EGFR (*Unni et al., 2015*).

## Discussion

The pattern of mutual exclusivity observed with mutant *EGFR* and mutant *KRAS* genes in LUAD is a consequence of synthetic lethality, not pathway redundancy; co-expression of these oncogenes is toxic, resulting in loss of viable cells (*Unni et al., 2015*; *Varmus et al., 2016*). There are reports of exceptions to this mutual exclusivity but these arise in conditions that include inhibition of EGFR (*Blakely et al., 2017*; *Ramalingam et al., 2018*). This is to be expected, as cells treated with kinase inhibitors are not experiencing the effects of both oncogenes (i.e. mutant EGFR and mutant KRAS). A cancer cell that has not been exposed to inhibitors (e.g. against mutant EGFR) could arise, particularly at an advanced stage of disease, with activating mutations in both EGFR and KRAS; but we would anticipate that other events—like decreased RAS-GTP levels—might prevent P-ERK from reaching toxic levels.

Despite the possible exceptions, it remains critical to understand why, based on the pattern of mutual exclusion, cells are generally unable to tolerate the combination of these two oncogenes more readily. And what are the biochemical mechanisms by which the toxicity is mediated, might be modulated to avoid lethality, or could be exploited therapeutically? To address these questions, we began by regulating the expression of mutant *KRAS* in LUAD cell lines carrying mutant *RAS* or *EGFR*

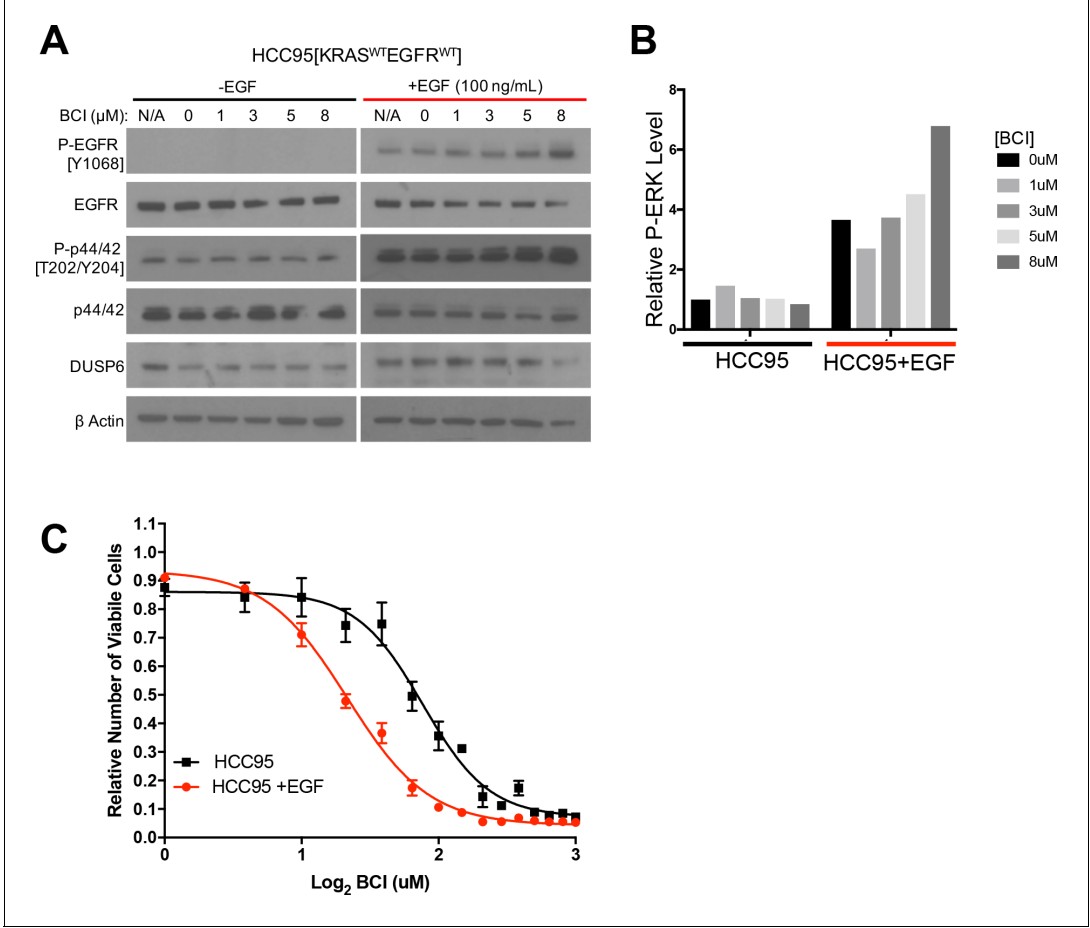

**Figure 5.** *EGF-mediated activation of ERK signaling leads to dependence on DUSP6.* (**A**) EGF increases P-ERK in HCC95 cells. BCI- insensitive HCC95 cells were grown in the presence and absence of EGF (100 ng/mL) and increasing doses of BCI; levels of the indicated proteins were assessed in cell lysates by Western blotting. EGF increased the levels of P-EGFR and P-ERK, and levels of P-ERK were further increased by BCI. (**B**) Relative P-ERK levels (ratio of phosphorylated to total levels normalized to actin) were determined by dosimetry and compared to the vehicle controls (0 BCI = 0.1% DMSO) to quantify the relative increase after BCI treatment from the gels in A. (**C**) Increase of P-ERK promotes sensitivity of lung cancer cell lines without *KRAS* or *EGFR* mutations to BCI. BCI- insensitive HCC95 cells were treated with 100 ng/mL of EGF for 10 days and then grown in medium containing escalating doses on BCI with continued EGF. Viable cells were measured 72 hr later with Alamar blue and compared to the vehicle controls (in 0.1% DMSO) to assess the relative change in numbers of viable cells. Experiments were done in biological triplicate with the average values presented ±SEM. The EGF-treated cells (red line) showed increased sensitivity (decreased viable cells at lower BCI conditions) than those without EGF treatment (black line). (**B–C**).

DOI: https://doi.org/10.7554/eLife.33718.010

The following figure supplement is available for figure 5:

**Figure supplement 1.** Protein lysates from conditions indicated in *Figure 5A* were subjected to electrophoresis on the same gel to directly compare p-EGFR and P-ERK levels in EGF-treated and untreated HCC95 cells.

DOI: https://doi.org/10.7554/eLife.33718.011

alleles. The levels of RAS activation in these cells are not expected to mirror what is found in tumors; these levels presumably will exceed what tumors can tolerate. We suggest that tumor cells could experience this state during progression, particularly when co-mutations in the RAS pathway have occurred. Understanding how the toxicity arises provides insight into mutual exclusivity and how limits for RAS activation may be set and exploited in cancer cells.

Our efforts to answer these questions have led to the conclusions that the toxicity is mediated through the hyperactivity of phosphorylated ERK1/2 and that inhibition of DUSP6 may re-create the toxicity through the role of this phosphatase as a negative regulator of ERK1/2. Several results reported here support these conclusions: (i) the previously reported toxicity that results from co-

expression of mutant *EGFR* and mutant *KRAS* is accompanied by an early increase in the phosphorylation of ERK1/2, and the effects can be attenuated by inhibiting MEK (which phosphorylates and activates ERK1/2) or by reducing ERK levels with inhibitory RNAs; (ii) DUSP6, a phosphatase known to be a feedback inhibitor of ERK activity, is present at relatively high levels in LUADs with *EGFR* and *KRAS* mutations; and (iii) inhibition of DUSP6, either by introduction of siRNAs or by treatment with the drug BCI, reduces the number of viable LUAD cells with *EGFR* or *KRAS* mutations or of BCI-resistant cells exposed to EGF.

Taken in concert, these findings support a general hypothesis about cell signaling. Activation of a biochemical signal from a critical node, such as ERK, in a signaling pathway must rise to a certain level to drive neoplastic changes in cell behavior; if signal intensity falls below that level, the cells may revert to a normal phenotype or initiate cell death as a manifestation of what is often called 'oncogene addiction" (*Nissan et al., 2013*; *Weinstein et al., 1997*; *Dow et al., 2015*; *Varmus et al., 2005*; *Sharma et al., 2006*). Conversely, if the intensity of signaling rises to exceed a higher threshold, the cells may display a variety of toxic effects, including senescence, vacuolization, or apoptosis (*Unni et al., 2015*; *Chi et al., 1999*; *Serrano et al., 1997*; *Joneson and Bar-Sagi, 1999*; *Overmeyer et al., 2008*; *Zhu et al., 1998*). In this model, two approaches to cancer therapy can be envisioned: (i) blocks to signaling that reverse the oncogenic phenotype or induce the apoptosis associated with oncogene addiction, or (ii) enhancements of signaling that cause selective toxicity in cells with pre-existing oncogenic mutations, a form of synthetic lethality that depends on changes that produce a gain rather than a loss of function. The former is exemplified by using inhibitors of EGFR kinase activity to induce remissions in LUAD with EGFR mutations (*Lynch et al., 2004*; *Paez et al., 2004*; *Pao et al., 2004*). Based on the findings presented here, the latter strategy might be pursued by using inhibitors of DUSP6 or other negative feedback regulators to block its usual attenuation of signals emanating from activated ERK1/2.

Several factors are likely to determine the threshold for producing the cell toxicity driven by hyperactive signaling nodes, such as ERKs, in cancer cells. These factors are likely to include allele-specific attributes of oncogenic mutations in genes such as *KRAS* (*Hunter et al., 2015*) and *BRAF* (*Hunter et al., 2015*; *Yao et al., 2017*; *Nieto et al., 2017*); the cell lineage in which the cancer has arisen (*Shojaee et al., 2015*; *Yao et al., 2017*; *Zhao et al., 2015*); the levels of expression of mutant cancer genes (*Zhu et al., 1998*; *Nieto et al., 2017*; *Cisowski et al., 2016*; *Ambrogio et al., 2017*); the co-existence of certain additional mutations (*Barretina et al., 2012*); and the multiple proteins that negatively regulate oncogenic proteins through feedback loops, such as MIG6 on EGFR (*Ambrogio et al., 2017*; *Maity et al., 2015*; *Anastasi et al., 2016*), GAPs on RAS proteins (*Courtois-Cox et al., 2006*; *Vigil et al., 2010*), or SPROUTYs and DUSPs on kinases downstream of RAS (*Kidger and Keyse, 2016*; *Shojaee et al., 2015*; *Zhao et al., 2015*). All such factors would need to be considered in the design of therapeutic strategies to generate signal intensities that are intolerable specifically in cancer cells. DUSP6 is a well-established negative regulator of ERK activation in a normal cellular context (reviewed in *Keyse, 2008*, and *Theodosiou and Ashworth, 2002*), so it is perhaps not surprising that this protein appears to have a critical role in persistently limiting ERK activation, even in a pathological context such as cancer.

The findings presented here, as well as recent results from others (*Shojaee et al., 2015*; *Leung et al., 2018*; *Wittig-Blaich et al., 2017*), support several underlying features of a therapeutic strategy based on inordinate signaling activity involving RAS proteins: that the activity of ERK needs to be actively controlled in cancer cells of diverse tissue origins; that hyperactivation of ERK can be deleterious to cells; and that inhibition of negative regulators like DUSP6 can create a toxic cellular state. This leads to the hypothesis that cancer cells *dependent* on ERK signaling have an active RTK-RAS-RAF-MEK pathway that produces levels of activated (phosphorylated) ERK1/2 that *require* attenuation. In other words, ERK-dependent tumor cells, including cancers driven by mutant RTK, RAS, BRAF, or MEK proteins, will have a vulnerability to hyperactivated ERK and that vulnerability can potentially be exploited by inhibition of feedback regulators like DUSP6.

Relevant to this concept are recent studies that address 'drug addiction' whereby cells lose viability when the inhibitor (e.g. vemurafenib) is removed (*Hong et al., 2018*; *Kong et al., 2017*; *Das Thakur et al., 2013*; *Moriceau et al., 2015*; *Sun et al., 2014*). These scenarios, in which an additional mutation can arise in the RTK-RAS-RAF-MEK pathway, create conditions similar to those we have modeled, once the inhibitor is removed. Additionally, Hata et al. have shown that mutations can arise while cells are exposed to a drug; as mentioned above, such mutations might appear to

violate patterns of mutual exclusivity but the pattern only arose because of pathway down-modulation (*Hata et al., 2016*) Recently, Leung et al. have found a similar dependency on ERK activation limits in mutant BRAF-driven melanoma (*Leung et al., 2018*).

The mechanisms of cell toxicity that arise from hyper-activation of ERK are likely to be diverse. We previously documented autophagy, apoptosis and macropinocytosis in cells expressing mutant EGFR and mutant KRAS, and others have described parthanatos and pseudosenescence as mechanisms for cell death from hyper-activation of ERK (*Hong et al., 2018*). ERK-dependent processes may differ from cell type to cell type based on mutation profiles and cellular state at the time of ERK activation. This same dependence on ERK (ERK2 specifically) has been documented for senescence when mutant RAS is introduced into normal cells (*Shin et al., 2013*).

The hypothesis that DUSP6 regulates ERK activity in the presence of signaling through the RAS pathway is particularly attractive in view of the frequency of *RAS* gene mutations in human cancers and the difficulties of targeting mutant RAS proteins (*Simanshu et al., 2017*; *Papke and Der, 2017*; *Downward, 2015*). Because DUSP6 directly controls the activities of ERK1 and ERK2, rather than proteins further upstream in the signaling pathway, it appears to be well-situated for controlling both the signal delivered to ERK through the activation of RAS and the signal emitted by phosphorylated ERK. Recently, Wittig-Blaich et al. have also found that inhibition of DUSP6 by siRNA was toxic in melanoma cells carrying mutant BRAF (*Wittig-Blaich et al., 2017*). Inhibition of other DUSPs, like DUSP5, that regulate ERK1 and ERK2 may create similar vulnerabilities and should be explored (*Kidger and Keyse, 2016*; *Kidger et al., 2017*). These ideas should provoke searches for inhibitors of DUSPs and other feedback inhibitors of this signaling pathway, as well as experiments that better define the downstream mediators and the consequences of non-attenuated ERK signaling.

# Materials and methods

## Cell lines and culture conditions

PC9 (PC-9), H358 (NCI-H358), H1975 (NCI-H1975), H1648 (NCI-H1648), A549, H460 (NCI-H460), H23 (NCI-H23), H2122 (NCI-H2122), H1650 (NCI-H1650), H2009 (NCI-H2009), H2030 (NCI-H2030), H1437 (NCI-H1437) and HCC95 cells were obtained from American Type Tissue Culture (ATCC) or were a kind gift from Dr. Adi Gazdar (UTSW) or Dr. Romel Somwar (MSKCC). Cell lines were periodically checked for mycoplasm contamination and found to be negative. Cells have been validated by STR profiling. For experiments involving doxycycline inducible constructs, cells were maintained in RPMI-1640 medium (Lonza) supplemented with 10% Tetracycline-free FBS (Clontech) or FBS that was tested to be Tet-free (VWR Life Science Seradigm), 10 mM HEPES (Gibco) and 1 mM Sodium pyruvate (Gibco). For other experiments, cells were grown in RPMI-1640 medium (Thermo Fisher) supplemented with 10% FBS (Sigma), 1% Glutamax (Thermo Fisher) and Pen/Strep (Thermo Fisher). Cells were cultured at 37°; air; 95%; CO2, 5%. Where indicated, doxycycline hyclate (Sigma-Aldrich) was added at the time of cell seeding at 100 ng/ml. Trametinib (Selleckchem), Buparlisib (Selleckchem), SCH772984 (Selleckchem), Dual Specificity protein phosphatase 1/6 inhibitor (BCI) (Calbiochem), and EGF recombinant human protein solution (Thermo Fisher) were added at the time of cell seeding at the indicated doses.

## Plasmids and generation of stable cell lines

Plasmids used were identical to those described in a prior publication (*Unni et al., 2015*). In brief, DNAs encoding mutant KRAS or GFP were cloned into pInducer20, a vector that carries a tetracycline response element for dox-dependent gene control and encodes rtTA, driven from the UbC promoter (*Meerbrey et al., 2011*). Lentivirus was generated using 293 T cells (ATCC), psPAX2 #12260 (Addgene, Cambridge, MA) and pMD2.G (Addgene plasmid#12259). Polyclonal cell lines (H358-tetO-GFP, H358-tetO-KRAS[G12V], PC9-tetO-GFP, H1975-tetO-GFP) and single cell-derived clonal cell lines (PC9-tetO-KRAS[G12V], H1975-tetO-KRAS[G12V]) were used. pLKO.1-based lentiviral vectors were used to establish cells stably expressing shRNAs for the indicated genes. Knockdown was achieved using two independent shRNAs targeting *ERK1* (noted in text as A4 or ERK1-4 and A5 or ERK1-5) or *ERK2* (noted in text as G6 or ERK2-6 and G7 or ERK2-7) RNAs. shRNA-GFP: GCAAGCTGACCCTGAAGTTCAT shRNA-ERK1 (A4): CGACCTTAAGATTTGTGATTT shRNA-ERK1 (A5): CTATACCAAGTCCATCGACAT shRNA-ERK2 (G6): TATTACGACCCGAGTGACGAG shRNA-ERK2 (G7):

TGGAATTGGATGACTTGCCTA shRNAs targeting GFP or a scramble sequence were used as controls. shRNA constructs were kindly provided by J. Blenis, Weill Cornell Medicine. Lentivirus was generated using 293 T cells as above. After transduction, polyclonal cells were selected with puromycin and maintained as a stable cell line.

## Measurements of protein levels

Cells were lysed in RIPA buffer (Boston Bioproducts) containing Halt protease and phosphatase inhibitor cocktail (Thermo Fisher). For experiments involving dox-inducible constructs, lysates were cleared by centrifugation, and protein concentration determined by Pierce BCA protein assay kit (Thermo Fisher). Samples were denatured by boiling in loading buffer (Cell Signaling). 20 µg of lysates were loaded on 10% MiniProtean TGX gels (Bio-Rad), transferred to Immun-Blot PVDF membranes (Bio-Rad), blocked in TBST (0.1% Tween-20) and 5% milk. For all other experiments, samples were denatured by boiling in loading buffer (BioRad) and 25 µg of lysates were loaded on 4–12% Bis-Tris gradient gels (Thermo Fisher), run using MOPS buffer, transferred to Immobilon-P PVDF membranes (Millipore) and blocked in TBST (0.1% Tween-20)/5% BSA (Sigma).

Primary incubation with antibodies was performed overnight at 4° in 5% BSA, followed by appropriate HRP-conjugated secondary antisera (Santa Cruz Biotechnology) and detected using ECL (Thermo Fisher). Antibodies were obtained from Cell Signaling and raised against the following proteins: phospho p-38 (4511), p38 (8690), p-p44/p42 (ERK1/2) (9101), p44/p42 (ERK1/2) (4695), p-SAPK/JNK (4668), SAPK/JNK (9252), P-EGFR (3777, 2234), EGFR (2232), KRAS (8955), PARP (9542), cleaved-PARP (5625), α-Tubulin (3873) and β-Actin (3700, 4970). Additionally, we used an antibody against GFP (A-21311, Thermo Fisher), DUSP1 (ab1351, abcam) and DUSP6 (ab76310, abcam and SC-377070, SC-137426, Santa Cruz)..

For 24 hr time course experiments, 100,000 cells (PC9, H1975) or 500,000 cells (H358) per well were seeded in a 6-well plate and stimulated with dox or dox and drug. For 5 day experiments, 25,000 cells were seeded in 6-well format. For 7 day time course experiments, 300,000 cells (H358) or 30,000 cells (H1975) were seeded into 10 cM plates and media was changed every day.

For proteome profiler array, 200 ug of total lysate was incubated on membranes in the A/B set (ARY003B, R and D Systems) and processed according to protocol (R and D Systems). Film exposures were scanned and spot density quantified using Image Studio Lite (Licor). Data were plotted in Microsoft Excel.

For western blots with BCI and Trametinib, cells were seeded to achieve 80% confluency 18 hr post seeding. Medium was aspirated and replaced with antibiotic-free medium containing drug at indicated concentrations and incubated for 30 min. Cells were lysed and protein levels assessed as stated above. Quantification of western blot images was performed using ImageJ software. Scanned files were saved in TIFF format, and background was subtracted from all images. Rectangle tool was used to fully encompass each separate band. Rectangles and bands were assigned lanes and histogram plots were generated based on each lane. Each histogram was enclosed using a straight line across the bottom and the 'magic wand' tool generated a value for area of histogram. These values were exported to and assessed using Excel and Graphpad Prism software.

## Measurements of viable cells

For experiments with dox-inducible constructs, cells were seeded into media containing doxycycline (100 ng/ml) and/or drug (Trametinib, SCH772984). Media (with or without doxycycline or drug) were replenished every 3 days during the 7 days. At indicated time points, medium was aspirated and replaced with medium containing Alamar Blue (Thermo Fisher). Fluorescence intensities from each well were read in duplicate on a FLUOstar Omega instrument (BMG Labtech), and data plotted in Microsoft Excel. Cells were seeded in triplicate in 24-well format at 1,000 cells/well (PC9 or H1975 derivatives) or 5,000 cells/well (H358 derivatives). For other experiments, cells were grown in 6-well plates, Alamar Blue added, and intensities measured for each well in quadruplicate using a Cytation 3 Multi Modal Reader with Gen5 software (BioTek).

For crystal violet assays, cells were seeded to achieve 80–90% confluency at the end point in the absence of drug treatment. 18 hr later, medium was aspirated and replaced with medium containing drug. Cells were incubated for 72 hr, washed with PBS and Crystal Violet solution (Sigma) was added and incubated for 2 min before washing again with PBS and imaging.

## Genomic datasets and analyses

RNA-Seq (RSEM) data for EGFR-KRAS-ERK pathway phosphatases (DUSP1-6, SPRED1-3, SPRY1-4) along with corresponding mutational data for *EGFR, KRAS, MET, ERBB2, BRAF, NF1, NRAS* and *HRAS* for 230 lung adenocarcinoma tumors from The Cancer Genome Atlas (*Cancer Genome Atlas Research Network, 2014*) were downloaded from cBioPortal (http://www.cbioportal.org/) (*Cerami et al., 2012*; *Gao et al., 2013*). Expression of each gene was compared between tumors with *KRAS* or *EGFR* mutations and those without, using an unpaired T-Test. Resulting p-values were adjusted for multiple comparisons using a Bonferroni correction and the –$Log_2$ value plotted as an indication of significance. Normalized expression values (sample gene value – median gene expression across all samples/row median absolute deviation) for each gene were also plotted using MORPHEUS software (https://software.broadinstitute.org/morpheus, Broad Institute) as a heat map. Expression of DUSP6 was also individually compared for tumors with EGFR mutation only, KRAS mutation only, or any RTK-RAS-ERK pathway mutation (EGFR, KRAS, MET, BRAF, ERBB2, NRAS, HRAS or NF1) vs those wild-type for the in each instance using a two-tailed Mann-Whitney U-Test in Prism 7 (Graphpad).

Reverse phase protein array (RPPA) data (replicate-base normalized [*Akbani et al., 2014*]) for 182/230 tumors were downloaded from the UCSC Cancer Genomics Browser. Levels of MAPKPT202Y204, P38PT180Y18 and JNKPT183Y185 were compared between samples with a *KRAS* or *EGFR* mutation and those without, using the Mann-Whitney U-Test.. Likewise, samples were separated into groups with high and low DUSP6 expression levels, based on the highest and lowest *DUSP6* expression quartiles; MAPKPT202Y204, P38PT180Y18 and JNKPT183Y185 levels were compared between the groups as above. Lastly, MAPKPT202Y204 levels from RPPA (RBN values) were correlated with *DUSP6* expression ($Log_2$ RSEM values), and the Pearson correlation coefficient and p-value determined. As phospho-protein levels were predicted to be higher in samples with KRAS or EGFR mutation or high DUSP6, one-tailed p-values were calculated.

*DUSP6* expression was also compared between tumors with and without *EGFR* or *KRAS* mutations in 83 tumors and matched normal lung tissues from the BC Cancer Agency (BCCA) and deposited in the Gene Expression Omnibus (GSE75037) as described above. Similarly, *DUSP6* expression was compared between human epithelial cells expressing various oncogenes or GFP control (GSE3151) (*Bild et al., 2006*). Lastly, Affymetrix Mouse Genome 430 2.0 Arrays were used to profile the lung from genetically engineered mouse models of lung cancer with and without the expression of different driver oncogenes (EGFR-DEL, EGFR-L858R, KRAS-G12D and MYC) (*Fisher et al., 2001*; *Politi et al., 2006*; *Podsypanina et al., 2008*) and levels of DUSP6 compared using a two-tailed Mann-Whitney U-Test in Prism seven software (Graphpad).

## siRNA transfections

For the time course experiments, 50,000 cells (PC9) per well were seeded in a 6-well plate. For the endpoint experiments, 50,000 cells (PC9, PC9-shERK1-5, PC9-shERK2-7, PC9-shScramble) or 75,000 cells (1975, A549, HCC95) per well were seeded. Cells were then transfected with ON-TARGETplus siRNA pools (Dharmacon) against the following targets as previously described (*Lockwood et al., 2012*)— EGFR (L-003114-00-0010), KIF11 (L-003317-00-0010), KRAS (L-005069-00-0010), DUSP6 (L-003964-00-0010)—as well as a non-targeting control (D-001810-10-20). In addition, to test specificity for DUSP6, siRNAs comprising the pool (J-003964-06-0005, J-003964-07-0005, J-003964-08-0005 and J-003964-09-0005) were also tested individually. An additional siRNA (Hs_DUSP6_6 FlexiTube siRNA SI03106404, Qiagen) targeting a different region of DUSP6 coding sequence than J-003964-08-0005 was tested to establish that the decreased viability was not due to off target effects.

DUSP6-8 (Dharmacon) Target Sequence: GGCATTAGCCGCTCAGTCA

DUSP6-Qiagen (Qiagen) Target Sequence: GTCGGAAATGGCGATCAGCAA

For consistent transfection efficiency across experiments, 10 uL of 20 uM siRNA pool was added in 190 uL of OptiMEM (Life Technologies) and 5 uL of Dharmafect was added in 195 uL of OptiMEM (Life Technologies) at room temperature. The siRNA and Dharmafect suspensions were mixed and incubated for 20 min prior to transfection. Media was changed 24 hr after transfection. For sustained knockdown of targets, transfections were conducted on Day 0 and again on Day 3. Viable cells were measured using Alamar Blue as described above. For the time course experiment, cell viability was determined on Day 1, Day 3 (prior to second transfection) and Day 5 or only on Day 5. Results were

compared between each siRNA and non-targeting control using a one-sample t-test as previously described (*Lockwood et al., 2012*).

## BCI dose-response treatments

Dose-response curves for BCI were established using a modified version of the protocol previously described (*Lockwood et al., 2012*). Briefly, cells were seeded in quadruplicate at optimal densities into 96-well plates containing media with and without BCI at indicated doses in 0.1% DMSO. Viable cells were measured 72 hr later with Alamar Blue as described above. All experiments were performed in at least biological duplicate and plotted ±SEM. For HCC95 sensitization assays, cells were cultured with or without 100 ng/mL of EGF Recombinant Human Protein Solution (Life Technologies) for 10 days prior to seeding in 96-well plates for BCI dose response assays with or without EGF. The cells were allowed to adhere for 24 hr before treatment with 17 different concentrations of BCI, ranging from 0 to 8 uM, with 0.5 uM increment doses at 0.1% DMSO concentration. Additionally, 100 uM of Etoposide (0.1% DMSO) was added as a positive control for cell death. Cell viability was determined after 72 hr of drug exposure using Alamar Blue. Graphpad Prism software was used to create dose response curves.

For BCI rescue experiments, 75,000 H358 cells were seeded in 6-well plates and adhered for 24 hr. After attachment, the cells were treated with varying combinations of VX-11e and BCI with the final DMSO concentration at 0.1% in each well. Cells were treated for 72 hr and then the media was switched with fresh media containing Alamar blue for viability assessment. Resulting values for each BCI + VX-11e containing well were normalized to well containing corresponding concentration of VX-11e only. Experiments were performed in biological triplicate and the average ±SEM plotted.

## Quantitative RT-PCR

Cells were homogenized and RNA extracted using the RNeasy Mini kit (Qiagen) according to the manufacturer's instructions. cDNA was prepared using the High-Capacity cDNA Reverse Transcription kit (Thermo Fisher). RT–PCR reactions were carried out using the TaqMan Gene Expression Master Mix (Thermo Fisher) and TaqMan Gene Expression Assays (Thermo Fischer) for *DUSP6* (Hs00169257_m1) and *GAPDH* (Hs99999905_m1). Reactions were run on a QuantStudio6 Real Time PCR system (Thermo Fisher). The ΔΔCt method was used for relative expression quantification using the average cycle thresholds.

## Genome-wide CRISPR screens

Genome-wide screens were performed with the Toronto Knockout version 3 (TKOv3) library (*Hart et al., 2017*). Lentivirus was generated from the TKOv3 library in low passage (<10) 293FT cells (Thermo Fisher) using Lipofectamine 3000 (Thermo Fisher). Approximately 120 million target cells were then infected with the TKOv3 library virus at an MOI of 0.3, in order to achieve an average 500-fold representation of the sgRNAs after selection. Cells were selected on puromycin for 7 days and then 35 million cells were seeded in culture. For the depletion screens, cells were passaged every 3 days, and after 14 population doublings, 35 million cells were harvested for genomic DNA extraction. For the enrichment screens, media (containing BCI or doxycycline) was changed every 3 days until cell death was no longer observed, at which point the remaining cells were harvested for genomic DNA extraction. sgRNA inserts were amplified with NEBNext High-Fidelity 2X PCR Master Mix (New England BioLabs). Samples were then purified and sequenced on a NextSeq 500 kit (Illumina).

For validation of the screen, two separate guides targeting KRAS were cloned into lentiCRISPR v2[75], lentivirus generated and H460 cells were transduced. Seven days after puromycin selection cells were harvested for protein analysis and seeded in the presence of BCI. A guide against LacZ was used as a control.

sgRNA_Lacz: GAGCGAACGCGTAACGCGAA  sgRNA_KRAS-1: GGACCAGTACATGAGGACTG sgRNA_KRAS-2: GTAGTTGGAGCTGGTGGCGT

For targeting of *DUSP6*, two separate guides were cloned into lentiCRISPR v2, lentivirus generated, and H358 cells were transduced. A clonal population of cells were expanded and screened by western blotting and by DNA sequencing of the *DUSP6* locus.

sgRNA_DUSP6-1:    GTGCGCGCGCTCTTCACGCG    sgRNA_DUSP6-2:    ACTCGTATAGCTCC TGCGGC

## Analysis of CRISPR screen

Sequencing reads were aligned to the reference library to determine the abundance of each sgRNA. sgRNAs with less than 30 raw read counts were excluded from further analysis. The read counts were then normalized to the total number of reads obtained from the respective sample. The log2 fold-change of each sgRNA was calculated by adding a pseudocount of 1 and comparing the abundance of the sgRNAs in the final cell population to their respective abundance in the TKOv3 plasmid library. Finally, genes were ranked according to the second-most enriched or second-most depleted sgRNA.

## Acknowledgements

We would like to thank Katerina Politi (Yale University) for providing gene expression data from her transgenic mice. We would like to thank members of the Varmus lab for useful discussions and Oksana Mashadova, in particular, for experimental help.

## Additional information

### Funding

| Funder | Grant reference number | Author |
|---|---|---|
| Canadian Institutes of Health Research | PJT-148725 | William W Lockwood |
| Terry Fox Research Institute | | William W Lockwood |
| Michael Smith Foundation for Health Research | Scholar Award | William W Lockwood |
| National Institutes of Health | | Harold Varmus |
| Meyer Cancer Center at Weill Cornell Medicine | | Harold Varmus |
| BC Cancer Foundation | | William W Lockwood |

The funders had no role in study design, data collection and interpretation, or the decision to submit the work for publication.

### Author contributions

Arun M Unni, William W Lockwood, Conceptualization, Data curation, Formal analysis, Supervision, Investigation, Methodology, Writing—original draft, Project administration, Writing—review and editing; Bryant Harbourne, Min Hee Oh, Data curation, Formal analysis, Investigation, Methodology; Sophia Wild, Data curation, Formal analysis; John R Ferrarone, Resources, Data curation, Formal analysis, Investigation; Harold Varmus, Conceptualization, Supervision, Writing—original draft, Project administration, Writing—review and editing

### Author ORCIDs

Arun M Unni (ID) http://orcid.org/0000-0003-0530-1470
William W Lockwood (ID) https://orcid.org/0000-0001-9831-3408

### Decision letter and Author response

Decision letter https://doi.org/10.7554/eLife.33718.021
Author response https://doi.org/10.7554/eLife.33718.022

## Additional files

### Supplementary files

• Supplementary file 1. Table containing the log2 fold change values for all sgRNAs from CRISPR-Cas9 screens.

DOI: https://doi.org/10.7554/eLife.33718.012

• Transparent reporting form

DOI: https://doi.org/10.7554/eLife.33718.013

### Data availability

All data generated or analysed during this study are included in the manuscript and supporting files. Source data files have been provided for Figures 2 and Figure 2-supplemental figure 1 in the Methods section and/or in the text.

The following previously published datasets were used:

| Author(s) | Year | Dataset title | Dataset URL | Database and Identifier |
|---|---|---|---|---|
| Cancer Genome Atlas Research Network | 2014 | TCGA LUAD | http://www.cbioportal.org/study?id=luad_tcga_pub#summary | cBioPortal, luad_tcga_pub |
| Gazdar A, Girard L, Stephen L, Wan L, Zhang W | 2017 | Expression profiling of 83 matched pairs of lung adenocarcinomas and non-malignant adjacent tissue | https://www.ncbi.nlm.nih.gov/geo/query/acc.cgi?acc=GSE75037 | NCBI Gene Expression Omnibus, GSE75037 |
| Nevins JR | 2005 | Oncogene Signature Dataset | https://www.ncbi.nlm.nih.gov/geo/query/acc.cgi?acc=GSE3151 | NCBI Gene Expression Omnibus, GSE3151 |

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
