## [Decision Letter]

[Editors’ note: formal revisions were requested, following approval of the authors’ plan of action.]

Thank you for submitting your article "Hyperactivation of ERK by mutation-driven RAS signaling or by inhibition of DUSP6 is toxic to lung adenocarcinoma cells" for consideration by *eLife*. Your article has been reviewed by three peer reviewers, one of whom (Thomas Look) is a member of our Board of Reviewing Editors, and Jonathan Cooper as the Senior Editor.

The reviewers have discussed the reviews with one another and the Reviewing Editor has drafted this letter to express the many issues that we feel must be addressed if this work is to advance to publication.

Summary:

In this manuscript Unni et al., describe their work on understanding the mechanism leading to cytotoxicity following forced overexpression of KRAS in LUAD cell lines harboring EGFR or KRAS mutations. They conclude that such toxicity owes to hyperactivation of pERK, which occurs through DUSP6. They show that MEK/ERK inhibitor treatment rescues the effect of forced overexpression while reducing the levels of pERK and that DUSP6 mRNA levels are increased in LUAD harboring EGFR or KRAS mutations relative to those with a wild type status of these proteins. Genetic or pharmacologic targeting of DUSP was found to induce pERK (transiently) and lead to diminished cell proliferation in EGFR/KRAS mutant cell lines but not in those with a wild type status. While overall the findings are intriguing, there are gaps in the experimental evidence provided which bias their conclusions. If the authors can address the issues noted below however this could be a very interesting article.

We have the following suggestions to improve the manuscript. For clarity, we address below the essential revisions related to the results presented in each figure.

1) Figure 1 a) Figure 1A: The expression level of tet-driven ectopic KRAS^G12V^ is very high in each of the three lung adenocarcinoma lines. The authors should demonstrate by western blotting whether these levels are comparable to endogenous KRAS levels found in lung adenocarcinoma lines. Studies from Tyler Jacks on mutant KRAS in knockin mice over 10 years ago emphasized the importance of expression levels of mutant KRAS to mediate transformation in lung adenocarcinoma. Massive overexpression of KRAS^G12V^ can induce senescence rather than transformation in most cell types.

b) Figure 1D: The trametinib/SCH984 experiments in Figure 1D require additional appropriate controls; the authors should show the effect of treatment in the absence of dox stimulation. Also, why are different concentrations of trametinib used?

c) In addition to the biochemistry and cell count data shown in each panel of Figure 1, straightforward studies need to be done to establish the cellular mechanism underlying the cell growth phenotype. Before and after induction of KRAS^G12V^ using dox, studies of cell cycle progression by DNA flow cytometry, for β-GAL expression assays should be done to assess senescence, and cleaved caspase 3, TUNEL and PARP cleavage assays should be done to assess apoptosis. The reduction of cell number 7-days after dox induction of KRAS^G12V^ of ~ 20-40 percent of control could be due to growth arrest, senescence or apoptosis. This will be important to establish with regard to the author's interpretation that over activation of ERK leads to synthetic lethality. The need for studies of cellular phenotype in the context of the experiments shown in Figure 1 apply broadly to experiments throughout the paper.

d) Does overexpression of KRASG12C in EGFR or KRAS WT cells also induce toxicity which can be mitigated by co-treatment with trametinib in cells with no activating mutations in EGFR-RAS-ERK pathway? The levels of overexpressed KRASG12C are so high that such levels could induce toxicity by exceeding the upper threshold of RAS-mediated signaling even in cells without hyperactive EGFR-KRAS-ERK signaling.

e) Since the main claim is that hyperactivation of ERK leads to toxicity, the authors ought to replicate their dox-inducible experiments with active ERK to show that these also lead to cytotoxicity. Also, just because inhibiting MEK/ERK reverses the some of the phenotype in plastic, this doesn't exclude that other RAS effectors are also involved (see induction of pAKT in addition to pERK in Figure 1B). The authors should thus carry out similar siRNA or inhibitor studies to demonstrate that the cytotoxic effects of KRAS overexpression are not due to pAKT.

2) Figure 2 a) Figure 2A: There should be symbols under the heat map denoting which tumors are KRAS mutant and which are EFGR mutant. Are there any tumors with ALK/ROS1/RET activation by translocation? Are there tumors with NF1, MET and BRAF mutations? These should be addressed since most of lung adenocarcinoma have activation of RTK/RAS/RAF pathway (TCGA, 2014) and increased dusp6 is among the five gene signatures to predict poor outcome (Chen et al., 2007).

b) After describing the emergence of DUSP6 as a key gene responding to activated ERK based on their own studies, the authors should put their findings of DUSP6 elevated expression in the context of previous work. It has been known for over 10 years that DUSP6 is transcriptionally regulated by ERK-responsive ETS transcription factors downstream of MAPK activation and that DUSP6 serves as a major negative feedback regulator of ERK signaling (Reffas et al., 2000; Kawakami et al., 2003; Eblaghie et al., 2003; Li et al., 2007; Ekerot et al., 2008; Furukawa et al., 2008; Jurek et al., 2009).

c) Figure 2B and C: For these panels, please show separate data categories for KRAS mutant and EGFR mutant tumors. The box plots shown do not really look statistically significantly different. The medians are close, and the quartiles are largely overlapping. Which statistical test was used to show significant differences? Has a biostatistician been consulted about whether parametric or non-parametric tests would be better for these comparisons? A detailed statistical section is needed in the Materials and methods section, outlining the statistical tests used for the data shown in each figure.

d) In Figure 2 A-C, the authors must also include the correlation between DUSP6 and P-p38 and p-JNK to show the readers that no correlation exists between them.

e) Figure 2D: Please provide information or citations about the mouse models used in this figure.

f) Figure 2E: The Material/Methods description about this experiment seems to be missing. Expression levels need to be shown of the transduced onco-proteins by Western blot to verify overexpression.

g) Figure 2G and H: Same comments as 2B and C.

3) Figure 3 a) Figure 3A: Only one siRNA was used for DUSP6 or EGFR and there is no rescue experiment to prove that the effect is due to knockdown of the intended target. DUSP6 de-phosphorylates ERK so ERK phosphorylation is expected to be increased with DUSP6 knockdown. Please explain why Figure 3A shows the opposite with reduced p-ERK after siDUSP6 knockdown.

b) Figure 3A: Can the toxicity mediated by DUSP6 knockdown in PC9 cells be reduced by co-treatment with trametinib (Figure 3A)? This would help confirm that toxicity caused by depletion of DUSP6 in these cells is due to increased pERK levels.

c) Figure 3B: PC9 cells in this panel show increased p-ERK after DUSP6 kd. Please explain why this is the opposite result compared to Panel A.

d) Figure 3B and C: Did the authors determine if the levels of DUSP6 in their EGFR and KRAS wild-type cell lines, HCC95 and H1648, are not as high as in the cell lines with EGFR or KRAS activating mutations? Lower levels of DUSP6 would indicate that these cells do not need to have similar buffering ability as the EGFR or KRAS mutant cell lines and would be consistent with their findings in Figure 2. In Figure 3B, it seems that DUSP6 is also buffering the pERK levels in HCC95 because knockdown of DUSP6 increases the levels of pERK in these cells by over 1.5 fold, similar to the cells with EGFR or KRAS activating mutations; but this increase in pERK has no impact on the cell survival (Figure 3C). This needs to be reconciled with the authors' main claim.

e) There seems to be a disconnect between the first part of the manuscript, where the authors describe the mechanism of toxicity caused by forced overexpression of KRAS and the second part, where they describe the effect of DUSP inhibition in cell lines with endogenous KRAS/EGFR mutations. To close this gap, the authors need to determine the levels of DUSP6 protein in their dox-inducible KRAS models and how they correlate with pERK. They should determine if the induction of pERK is transient, as that observed with DUSP6 siRNA, or sustained. Finally, they should use DUSP6 siRNA or BCI to determine if this reverses the effect of forced KRAS expression on pERK or proliferation. Another question that has not been addressed is why doesn't forced overexpression of KRAS induce sufficient DUSP6 to override the induction of pERK, if DUSP expression is under the control of EGFR/KRAS? Could there be other factors involved? If the authors can experimentally address these it would be a significant improvement.

f) More evidence is needed to show that the increase in DUSP mRNA is associated with an increased in DUSP6 protein levels or increased DUSP activity. The data in Figure 2I, where the authors compare the relationship between pERK (measured by RPPA) and DUSP6 mRNA is the only such evidence provided. This is underwhelming because the correlation is not great (r=0.1) and because the pERK does not seem to have been controlled for total ERK.

g) Finding that DUSP6 siRNA caused only a transient elevation in pERK (followed by inhibition of pERK at longer intervals), while mimicking the antiproliferative effect observed with forced overexpression of KRAS needs to be clarified with additional experiments. As it stands, it is difficult to agree with the authors conclusion that DUSP6 is the main mediator of the proliferative effect or that the effect of DUSP6 is through pERK. One consideration may be to use constitutively active ERK (i.e. DD phosphomimetic mutants) and attempt to reverse the effect of DUSP6. This is another point that, if addressed experimentally, could add value to the manuscript.

h) If indeed it is true that a transient induction in pERK leads to cytotoxicity several days later, then does EGF stimulation (which also causes a transient induction in pERK) have the same effect in EGFR WT/KRAS MT cells?

i) All three reviewers noted that HCC95 (RRID:CVCL_5137) is a squamous cell carcinoma line. This not comparable to lung adenocarcinoma because this tumor type has different pathway dependencies than LUAD (TCGA, Nature 2012). A wild type LUAD line should be tested instead.

j) Figure 3C: Same comments as 3A.

k) The investigators might consider inactivating DUSP6 with CRISPR-Cas9 in these cell lines to show genetic dependence.

4) Figure 4 a) Figure 4A: 11 lung cancer cell lines were treated with BCI and viable cells were measured after 72-hour treatment. Most of these cell lines are fast growing cells with doubling time around 20~30 hours. Reduction of viable cells by 72-hours does not necessarily mean cell killing. TUNEL or caspase-3/PARP western blot needs to be performed to detect the levels of apoptosis. Senescence and cell cycle arrest should be examined too.

b) Figure 4C and D: p-ERK levels at a single time point was measured in multiple cell lines (30 min). In order to explain the loss of viable cells at 72hours in Figure 4A, p-ERK should be measured at serial time points such as 1, 6, 12, 24, 48, 72 hours.

c) The data on BCI are very interesting but it's not clear how have the authors established that BCI is selective for DUSP? Sensitive and insensitive cell lines all have IC50s in the 1-5μM range (12/13 cells lines tested). What is the IC50 of the inhibitor in DUSP6 KD or KO cells (or in DUSP1/6 double KD/KO cells)? How do the authors know that the antiproliferative effect of BCI is due to its ability to induce pERK? These questions need to be addressed experimentally. The authors should also attempt to show that the inhibitor has an effect in vivo, given their claim that such an approach could be therapeutically beneficial in patients.

d) Figure 4E and F: Four cell lines were treated with 125nM trametinib for 72 hours. 125nM is a very high dose for trametinib-sensitive cell lines such as H358 (reported IC50 ~50nM). Apoptosis levels should be measured to document any cell death.

5) Figure 5.

See comments for Figure 3B. HCC95 is a SQCC cell line that is quite different from the LUAD cells used in the rest of this manuscript.

a) In figures where the authors make a comparison of protein levels under different conditions the uncut blot with all the lanes on the same blot must be shown. It is not ideal to make a comparison between levels of proteins on two different strips of blots. For example, in Figure 5B, authors claim that EGF increase levels of pEGFR and pERK in HCC95 cells when the pEGFR and pERK levels with and without EGF are on two different strips of blots. This is important to confirm that BCI treatment increases pERK levels in EGFR treated cells. Quantifications should be shown in addition.

Overall comment:

The authors should soften earlier claims that EGFR-mutations and KRAS-mutations are synthetically lethal in the adenocarcinoma subtype of NSCLC (Unni et al., 2015). Recent genetic studies of EGFR-mutant lung cancer (Blakely et al., 2017) have shown that 2.5%~4.7% EGFR-mutant lung cancer also harbor KRAS copy number gain or activating mutations. Tumor genomic analyses have indicated that bona fide driver mutations causing lung adenocarcinoma are not as mutually exclusive as previously thought (e.g. PMIDs: 25301630, 28498782, 28445112, 29106415, and other recent publications), particularly in metastatic lung adenocarcinoma instead of early-stage disease and/or during the evolution of treatment resistance in metastatic disease. Further, preclinical studies have shown acquired KRAS gain/activation in response to EGFRi in EGFR-mutant NSCLC, indicating that this can be a mechanism of drug resistance (Politi et al., 2000; Eberlein et al., 2015). The authors should acknowledge these recent works and temper their claim of absolute mutually exclusivity in this disease, at least under these more advanced-stage disease contexts and in the light of the emerging literature.

[Editors' note: further revisions were requested prior to acceptance, as described below.]

Thank you for resubmitting your work entitled "Hyperactivation of ERK by mutation-driven RAS signaling or by inhibition of DUSP6 is toxic to lung adenocarcinoma cells" for further consideration at *eLife*. First, let me apologize profusely for the very long time this has taken. The problem is basically that the reviewers are at an impasse and could not decide on a course of action.

Briefly, this is a follow up to a previous paper, that reported anti-proliferative effects of activation of Ras and EGFR in the same cancer cells. This new paper has provides evidence that the anti-proliferative effect of over-expressed, active Ras is due to ERK hyperactivation, and that DUSP6 is critical to prevent adverse effects of active ERK in lung cancer cells. In the revision, the authors have added results using ERK shRNA, ERKi, and a CRISPR screen that together provide convincing evidence that ERK mediates the antiproliferative effect of oncogene overexpression. The finding that sgKRAS are enriched in the BCI screen also indirectly supports that conclusion. The authors have also ruled out that the PI3K pathway is not involved by adding PI3K inhibitor experiments. These additions greatly strengthen the conclusion that hyperactivation of ERK is detrimental. However, the evidence that DUSP6 plays a key role is not convincing.

The DUSP6 knockdown was done with a single siRNA with no rescue experiment. (Other siRNAs did not effectively inhibit DUSP6 expression). Attempts to recapitulate the result with a DUSP6 CRISPR knockout were unsuccessful. Therefore, the conclusion that DUSP6 is necessary relies in large part on the specificity of the chemical BCI. The DUSP6 KO cells have the same IC50 to BCI, while DUSP1 siRNA did not affect proliferation in their system. Based on these results, the authors cannot claim that the BCI effect is DUSP6 specific nor that the BCI effect in DUSP6KO cells is driven by DUSP1 inhibition. It probably isn't, and by inference, the observed phenotype is not dependent on DUSP6 alone but on other ERK specific DUSPs as well.

The reviewers disagreed whether these weaknesses undermined the impact to the point where the paper was unsuitable for publication, or whether lengthy additional experiments would be needed. The *eLife* approach is not to ask for multiple rounds of revision. In this spirit, I suggest two possibilities:

Either

- Provide convincing evidence that validates DUSP6 as the key enzyme that downregulates ERK in lung cancer cells (e.g. try more siRNAs to find another that gives strong knockdown, and/or rescue DUSP6 siRNA with DUSP6 from mouse, or with silent mutations in the siRNA target sequence).

Or,

- Modify the title and abstract to allow for the possibility that other DUSPs are involved and be more open about the shortcomings of the results.

---

## [Author Response]

[Editors’ notes: the authors’ response after being formally invited to submit a revised submission follows.]

Summary:In this manuscript Unni et al., describe their work on understanding the mechanism leading to cytotoxicity following forced overexpression of KRAS in LUAD cell lines harboring EGFR or KRAS mutations. They conclude that such toxicity owes to hyperactivation of pERK, which occurs through DUSP6. They show that MEK/ERK inhibitor treatment rescues the effect of forced overexpression while reducing the levels of pERK and that DUSP6 mRNA levels are increased in LUAD harboring EGFR or KRAS mutations relative to those with a wild type status of these proteins. Genetic or pharmacologic targeting of DUSP was found to induce pERK (transiently) and lead to diminished cell proliferation in EGFR/KRAS mutant cell lines but not in those with a wild type status. While overall the findings are intriguing, there are gaps in the experimental evidence provided which bias their conclusions. If the authors can address the issues noted below however this could be a very interesting article.

The main message of our paper is that p-ERK hyperactivation is intolerable in cancer cells and that this property—the toxic consequences of exceeding a certain level of activated (phosphorylated) ERK—creates a therapeutic target in RAS pathway-mutated cancers: DUSP6, an ERK phosphatase that plays a major role in modulating the activity of ERK in lung cancer cells.

We arrived at these findings by studying the mutually exclusive pattern of EGFR and KRAS mutations in lung adenocarcinoma. As your review points out, there may be exceptions to this mutual exclusivity, but we do not believe they undermine our arguments. Nevertheless, we will make changes to the Discussion section to include the observations that the reviewers cite and also note their shortcomings (such as the difficulty of knowing whether normally excluded combinations of mutations have occurred in the same cell or separately in tumor subclones). Of particular relevance, p-ERK intolerance has also been documented in ‘drug addicted’ cells when inhibitors are removed (Kong et al., 2017; Hong et al., 2018, Das Thakur et al., 2013, Moriceau et al., 2015, Sun et al., 2014). Cells in these conditions appear to violate the mutual exclusivity pattern, however, we would argue that these mutations could have arisen while cells were on drug (Hata et al., 2016).

We have the following suggestions to improve the manuscript. For clarity, we address below the essential revisions related to the results presented in each figure.1) Figure 1 a) Figure 1A: The expression level of tet-driven ectopic KRAS^G12V^ is very high in each of the three lung adenocarcinoma lines. The authors should demonstrate by western blotting whether these levels are comparable to endogenous KRAS levels found in lung adenocarcinoma lines. Studies from Tyler Jacks on mutant KRAS in knockin mice over 10 years ago emphasized the importance of expression levels of mutant KRAS to mediate transformation in lung adenocarcinoma. Massive overexpression of KRAS^G12V^ can induce senescence rather than transformation in most cell types.

Our intention was to force excess RAS pathway activation (beyond what is present in tumor cells) and determine if this is tolerated. The purpose of doing this was to model what might be happening when co-mutations arise in the RAS pathway. We recognize that these levels of RAS may not be commonly experienced by tumor cells and will state this explicitly in the text. Even though our system, like many others, is artificial, it provides an experimental platform for understanding why hyper activation of ERK occurs and allowed us to define a vulnerability (DUSP6 inhibition) that we could exploit.

We have now stated this in the text (Results section).

b) Figure 1D: The trametinib/SCH984 experiments in Figure 1D require additional appropriate controls; the authors should show the effect of treatment in the absence of dox stimulation. Also, why are different concentrations of trametinib used?

Different concentrations were required to rescue the phenotype in different cells. We will now provide a plot to show dose response (plus/minus dox and plus/minus drug). The degree to which p-ERK is induced in each cell line appeared to require different concentrations of a MEK inhibitor to reset p-ERK back to an acceptable level.

A dose response curve for doxycycline plus trametinib is now plotted and shown in Figure 1—figure supplement 1C.

c) In addition to the biochemistry and cell count data shown in each panel of Figure 1, straightforward studies need to be done to establish the cellular mechanism underlying the cell growth phenotype. Before and after induction of KRAS^G12V^ using dox, studies of cell cycle progression by DNA flow cytometry, for β-GAL expression assays should be done to assess senescence, and cleaved caspase 3, TUNEL and PARP cleavage assays should be done to assess apoptosis. The reduction of cell number 7-days after dox induction of KRAS^G12V^ of ~ 20-40 percent of control could be due to growth arrest, senescence or apoptosis. This will be important to establish with regard to the author's interpretation that over activation of ERK leads to synthetic lethality. The need for studies of cellular phenotype in the context of the experiments shown in Figure 1 apply broadly to experiments throughout the paper.

We have previously published some of the effects of co-induction of mutant EGFR and mutant KRAS. In that study, we documented apoptosis, autophagy, vacuolization and macropinocytosis in cell lines similar to those we now use (Unni et al., 2015). A recent study (Hong et al., 2018) found that cancer cells that were ‘drug addicted’ die by apoptosis, parthanatos and pseudosenescence when inhibitors were removed (similar p-ERK overload principle). For these reasons there are likely to be several distinct mechanisms that result in a loss of cell viability. Our goal here has been to focus on the factors that generate the cytotoxic signal, rather than on a cell’s response to the signal. Nevertheless, in response to the reasonable concerns raised by this comment, we will assess the extent of apoptosis by measuring cleaved PARP, CASP3 activity, or Annexin V levels to help clarify our statements about cell toxicity.

We have measured cleaved PARP in the H358 and H1975 experimental systems described in this manuscript and some assays are shown in Figure 1—figure supplement 1. We characterized mechanisms of cell toxicity in PC9 cells that express both mutant EGFR and mutant KRAS in our earlier paper in *eLife*.

d) Does overexpression of KRASG12C in EGFR or KRAS WT cells also induce toxicity which can be mitigated by co-treatment with trametinib in cells with no activating mutations in EGFR-RAS-ERK pathway? The levels of overexpressed KRASG12C are so high that such levels could induce toxicity by exceeding the upper threshold of RAS-mediated signaling even in cells without hyperactive EGFR-KRAS-ERK signaling.

Overexpression of KRAS G12V is likely to result in cell death or senescence in a variety of cell lines, as others have shown in the past. However, the motivation for experiments in Figure 1 was to test the limits of *cancer* cells to activation of the RAS pathway in a reproducible way that could allow us to study the mechanism by which the toxic signals arise.

We have added a comment on RAS-mediated senescence in the text (Discussion section).

e) Since the main claim is that hyperactivation of ERK leads to toxicity, the authors ought to replicate their dox-inducible experiments with active ERK to show that these also lead to cytotoxicity. Also, just because inhibiting MEK/ERK reverses the some of the phenotype in plastic, this doesn't exclude that other RAS effectors are also involved (see induction of pAKT in addition to pERK in Figure 1B). The authors should thus carry out similar siRNA or inhibitor studies to demonstrate that the cytotoxic effects of KRAS overexpression are not due to pAKT.

Including data that an inducible active ERK2 allele is toxic would be valuable. However, creating and characterizing these lines will take a significant amount of time. We have prioritized other experiments that the reviewers advise that will strengthen our paper. We did show increases in p-AKT at an early time point and it is possible that effectors of RAS other than RAF proteins can also be toxic to cells. To address this point, tetO KRAS G12V cells will be treated with a PI3K inhibitor (to inhibit AKT phosphorylation) and placed on dox. We will document the effects on cell viability, and on p-AKT and p-ERK signaling.

We have included data with a PI3K inhibitor (buparilisib) in H358-tetO-KRAS cells (Figure—figure supplement 1D). Using the same cell line, a genome wide CRISPR-Cas9 screen did not reveal an enrichment of guide RNA targeting PIK3CA (Figure 1—figure supplement 1F and Supplementary file 1).

2) Figure 2a) Figure 2A: There should be symbols under the heat map denoting which tumors are KRAS mutant and which are EFGR mutant. Are there any tumors with ALK/ROS1/RET activation by translocation? Are there tumors with NF1, MET and BRAF mutations? These should be addressed since most of lung adenocarcinoma have activation of RTK/RAS/RAF pathway (TCGA, Nature 2014) and increased dusp6 is among the five gene signatures to predict poor outcome (Chen et al., 2007).

Symbols will be added for mutant KRAS, EGFR, BRAF, NF1, MET, ERBB2 etc. Translocations were not assessed in this data set. The Chen et al., five gene signatures is interesting (*DUSP6, MMD, STAT1, ERBB3* and *LCK*). Perhaps this signature is most common in KRAS mutant tumors, something Chen et al., do not address. We will comment on these observations in the revised manuscript.

We have now indicated which tumors are KRAS mutants and which are EGFR mutants in Figure 2A. In addition, we have added another heat map with NF1, BRAF, MET, ERBB2, NRAS and HRAS status indicated as Figure 2—figure supplement 1A. Further, we have compared levels of DUSP6 mRNA among all tumors with RTK-RAS-RAF pathway mutations vs those with only wild type components in this pathway; see Figure—figure supplement 1C.

b) After describing the emergence of DUSP6 as a key gene responding to activated ERK based on their own studies, the authors should put their findings of DUSP6 elevated expression in the context of previous work. It has been known for over 10 years that DUSP6 is transcriptionally regulated by ERK-responsive ETS transcription factors downstream of MAPK activation and that DUSP6 serves as a major negative feedback regulator of ERK signaling (Reffas et al., 2000; Kawakami et al., 2003; Eblaghie et al., 2003; Li et al., 2007; Ekerot et al., 2008; Furukawa et al., 2008; Jurek et al., 2009).

The reviewers correctly point to many papers that highlight the significance of DUSP6 in controlling ERK activity. Our main point in this analysis was to discover which of the prominent negative regulators (DUSPs, SPRYs and SPREDs) have been significantly modulated in lung adenocarcinoma. This revealed that lung adenocarcinoma with mutations in KRAS or EGFR appear to rely on DUSP6 to actively restrain the RTK-RAS-RAF-MEK-ERK pathway. Our work also emphasizes that tumor cells have a level of ERK activation that is *still* subject to negative feedback regulation and that this reliance is a vulnerability based on the data we show in Figure 1. We will mention several of the papers the reviewers cite to properly document the previous work with DUSP6.

The papers suggested by the reviewers are cited through two reviews in the text (Discussion section).

c) Figure 2B and C: For these panels, please show separate data categories for KRAS mutant and EGFR mutant tumors. The box plots shown do not really look statistically significantly different. The medians are close, and the quartiles are largely overlapping. Which statistical test was used to show significant differences? Has a biostatistician been consulted about whether parametric or non-parametric tests would be better for these comparisons? A detailed statistical section is needed in the Materials and methods section, outlining the statistical tests used for the data shown in each figure.

We will include a detailed analysis of the statistical tests used and the rationale. We will also separate KRAS and EGFR mutant cases. However, it should be noted that assessing KRAS and EGFR mutant tumors in separate groups will limit sample numbers. As a result, statistical power will be reduced, especially in RPPA assessment where the number of samples available is already limiting. It was for this reason that samples with KRAS and EGFR mutations were pooled. Of note, the box plots in Figure 2B and C are on a log scale, suggesting that the differences seen between the medians are quite large.

We have now included more details about the statistical tests used in the methods section. We have also separated EGFR and KRAS mutant tumors and compared each to KRAS/EGFR WT groups and included these data in Figure 2—figure supplement 1B,D).

d) In Figure 2 A-C, the authors must also include the correlation between DUSP6 and P-p38 and p-JNK to show the readers that no correlation exists between them.

These data are part of 2H. No correlations were observed. We will also provide plots similar to 2I for p-JNK and p-p38.

Correlation plots for P-p38 and P-JNK have been included as Figure 2—figure supplement 1E,F.

e) Figure 2D: Please provide information or citations about the mouse models used in this figure.

We will provide the proper citations of the mouse models used (Politi et al., 2006; Fisher et al., 2001; Felsher and Bishop, 1999).

The appropriate citations for the mouse models are now included (subsection “DUSP6 is a major regulator of negative feedback, expressed in LUAD cells, and associated with KRAS and EGFR mutations and with high P-ERK levels”).

f) Figure 2E: The Material/Methods description about this experiment seems to be missing. Expression levels need to be shown of the transduced onco-proteins by Western blot to verify overexpression.

These data were retrieved from a previous publication and not adequately cited (GSE3151, Bild et al., 2006; Kim et al., 2010). We will correct this in the text to clearly state that it is from publicly available data.

This citation has been added, and the origin of the data is now clearly stated in the text (subsection “DUSP6 is a major regulator of negative feedback, expressed in LUAD cells, and associated with KRAS and EGFR mutations and with high P-ERK levels”).

g) Figure 2G and H: Same comments as 2B and C

EGFR and KRAS mutant tumors will be assessed separately. However, as mentioned above, combining these genotypes provides greater statistical power.

Addressed above and in Figure 2—figure supplement 1.

3) Figure 3 a) Figure 3A: Only one siRNA was used for DUSP6 or EGFR and there is no rescue experiment to prove that the effect is due to knockdown of the intended target. DUSP6 de-phosphorylates ERK so ERK phosphorylation is expected to be increased with DUSP6 knockdown. Please explain why Figure 3A shows the opposite with reduced p-ERK after siDUSP6 knockdown.

We have used pooled siRNA in the knockdown experiments. We now have PC9 cells expressing wildtype or catalytically inactive DUSP6. These cells will be used to verify that our DUSP6 siRNA is on-target (using siRNA against the 3’UTR of the endogenous *DUSP6* which is not represented in the transgenes).

Unfortunately, despite repeated efforts, no siRNAs targeting the 3’UTR proved to be effective at knocking down DUSP6. In addition to using the original siRNA pool, we have now transfected PC9 cells with each individual siRNA comprising the pool, four in total, all of which target the coding region. This revealed a dose-dependent effect of knockdown on growth inhibition: suppression but incomplete knockdown stimulated cell growth, whereas more complete inhibition was toxic to cells (Figure 3—figure supplement 1A,B). This conforms with our hypothesis and suggests the siRNAs for DUSP6 are on target.

The challenging aspect of this study was that knockdown of DUSP6 will result in increased p-ERK leading to increased DUSP6 mRNA, which is being targeted by the siRNA (‘technical’ feedback loop). p-ERK is reduced in this figure probably because the measurement was made on day 5 (see 3B) when the cells are dying. We will include a measure of apoptosis (cleaved PARP) at this time point. We will also provide a longer exposure of the western blot showing the efficiency of DUSP6 knockdown as there could be remaining DUSP6 protein. This may contribute to the decreased p-ERK on day 5.

We have assessed cleaved PARP on day 5 after DUSP6 knockdown and shown that it is indeed induced in EGFR mutant H1975 cells but not EGFR/KRAS wild-type HCC95 cells (Figure 3—figure supplement 1C). We conclude that P-ERK levels are low at day 5 after DUSP6 knockdown because cells with increased P-ERK have already become non-viable by this time. This point was further investigated in BCI experiments below.

b) Figure 3A: Can the toxicity mediated by DUSP6 knockdown in PC9 cells be reduced by co-treatment with trametinib (Figure 3A)? This would help confirm that toxicity caused by depletion of DUSP6 in these cells is due to increased pERK levels.

This experiment was tried several times, but we could not find a dose of Trametinib that rescues lethality. MEK and ERK inhibitors are lethal in this line (PC9) and this presents a technical problem: both ERK inhibition and ERK hyperactivation are not tolerated. However, H1975 cells are much more tolerant to ERK inhibition using SCH772984. Experiments are underway to treat H1975 cells that have received siRNA against DUSP6 with an ERK inhibitor like SCH772984 to try and rescue the loss of cell viability.

Addition of drug (SCH772984) to cells transfected with siRNA against DUSP6 led to indiscriminate toxicity; cells could not withstand the stress of transfection coupled with an ERK inhibitor. As an alternative strategy, we knocked down DUSP6 with siRNA in PC9 cells co-transduced with shRNAs targeting either ERK1 or ERK2 as used in Figure 1. Stable, viable cells were established with reduced ERK levels. These cells displayed increased relative viability after DUSP6 knockdown compared to shScramble control cells, further suggesting that ERK plays a role in mediating the toxic effects of DUSP6 inhibition (now shown in Figure 3—figure supplement 1G,H,I).

c) Figure 3B: PC9 cells in this panel show increased p-ERK after DUSP6 kd. Please explain why this is the opposite result compared to Panel A.

These data are from 24hr samples, not 5 day samples (Figure 3A). We expect that acute loss of DUSP6 should increase levels of p-ERK that will then be part of a feedback loop of *DUSP6* activation, followed by de-phosphorylation of ERK.

As mentioned above, 24 hours after DUSP6 knockdown, cells are still viable and P-ERK is induced whereas, at day 5, cells have induced cleaved PARP and demonstrate substantially decreased viability. We postulate that cells transfected with siRNA for DUSP6 develop high levels of P-ERK and subsequent cell death, leaving only cells with lower P-ERK at day 5.

d) Figure 3B and C: Did the authors determine if the levels of DUSP6 in their EGFR and KRAS wild-type cell lines, HCC95 and H1648, are not as high as in the cell lines with EGFR or KRAS activating mutations? Lower levels of DUSP6 would indicate that these cells do not need to have similar buffering ability as the EGFR or KRAS mutant cell lines and would be consistent with their findings in Figure 2. In Figure 3B, it seems that DUSP6 is also buffering the pERK levels in HCC95 because knockdown of DUSP6 increases the levels of pERK in these cells by over 1.5 fold, similar to the cells with EGFR or KRAS activating mutations; but this increase in pERK has no impact on the cell survival (Figure 3C). This needs to be reconciled with the authors' main claim.

We will provide the basal levels of DUSP6 across the lines in one figure. Additionally, we will include the quantitation of p-ERK changes from our independent experiments to help establish the fold changes.

We have included immunoblots indicating the basal levels of DUSP6 and P-ERK across the panel of cell lines (Figure 3—figure supplement 1D). Due to the variability associated with transfection-based experiments, we have now compiled dosimetry for three independent western blots in Figure 3B and plotted the results. This revealed that EGFR or KRAS mutant, but not wild type, cells consistently demonstrate increased P-ERK upon DUSP6 knockdown. All these results and their potential implications are now described in the text (subsection “Knockdown of DUSP6 elevates P-ERK and reduces viability of LUAD cells with either KRAS or EGFR oncogenic mutations”).

e) There seems to be a disconnect between the first part of the manuscript, where the authors describe the mechanism of toxicity caused by forced overexpression of KRAS and the second part, where they describe the effect of DUSP inhibition in cell lines with endogenous KRAS/EGFR mutations. To close this gap, the authors need to determine the levels of DUSP6 protein in their dox-inducible KRAS models and how they correlate with pERK. They should determine if the induction of pERK is transient, as that observed with DUSP6 siRNA, or sustained. Finally, they should use DUSP6 siRNA or BCI to determine if this reverses the effect of forced KRAS expression on pERK or proliferation. Another question that has not been addressed is why doesn't forced overexpression of KRAS induce sufficient DUSP6 to override the induction of pERK, if DUSP expression is under the control of EGFR/KRAS? Could there be other factors involved? If the authors can experimentally address these it would be a significant improvement.

We don’t understand why the reviewers believe there is a “disconnect” between the early and late phases of the manuscript. In fact, we believe that there is a logical flow from identification of p-ERK as the locus that transmits a toxic signal to the implication of DUSP6 as a critical regulator of the activity of ERK. So the notion of an informational gap is not clear to us. Nevertheless, we agree that the situation is complicated by the kinetics of activation and de-activation of the components of the signaling system, and we will follow the request to obtain more kinetic data. We will perform time course experiments in tetO-RAS lines, measuring p-ERK induction and DUSP6 protein levels at 1,3,5 and 7 days. Contrary to the reviewers’ speculation, we would not anticipate that DUSP6 siRNA or BCI would reverse the effects of forced KRAS expression; they should potentiate the effects of KRAS. On the other hand, it is unclear why DUSP6 cannot ‘override’ the induction of p-ERK. It is possible that p-ERK has fully localized to the nucleus and is no longer accessible to DUSP6. We will consider these possibilities in the revised manuscript.

The kinetics of p-ERK induction have been provided for H358 (days 1, 3, 5 and 7) and H1975 (day 7) cells in Figure 1—figure supplement 1B). The time course of induction of p-ERK upon treatment of H358 cells with BCI (at 1, 6, 12, 24, 48, 72 hours) is provided in Figure 4—figure supplement 1D. Experiments with BCI provide initial assessment of the kinetics of induction of p-ERK upon DUSP6 inactivation. The kinetics suggest that p-ERK induction for at least 24 hours is required before markers of apoptosis (PARP cleavage) are detected. Similar kinetics of p-ERK induction (at least 24-48 hours) before PARP cleavage detection were observed for H358-tetO-KRAS cells (subsection “Synthetic lethality induced by co-expression of mutant KRAS and EGFR is mediated through increased ERK Signalling”, subsection “P-ERK levels increase in LUAD cells after inhibition of DUSP6 by BCI, and P-ERK is required for BCI-mediated

toxicity.”).

f) More evidence is needed to show that the increase in DUSP mRNA is associated with an increased in DUSP6 protein levels or increased DUSP activity. The data in Figure 2I, where the authors compare the relationship between pERK (measured by RPPA) and DUSP6 mRNA is the only such evidence provided. This is underwhelming because the correlation is not great (r=0.1) and because the pERK does not seem to have been controlled for total ERK.

The RPPA assays (from TCGA) are controlled for total protein. The time course studies in tetO lines and BCI-treated cell lines may help address the issue of ‘sufficient’ levels of DUSP6.

We have performed and included in the manuscript time course experiments for both dox treatment in TetO cell lines and BCI treated cells as described above.

g) Finding that DUSP6 siRNA caused only a transient elevation in pERK (followed by inhibition of pERK at longer intervals), while mimicking the antiproliferative effect observed with forced overexpression of KRAS needs to be clarified with additional experiments. As it stands, it is difficult to agree with the authors conclusion that DUSP6 is the main mediator of the proliferative effect or that the effect of DUSP6 is through pERK. One consideration may be to use constitutively active ERK (i.e. DD phosphomimetic mutants) and attempt to reverse the effect of DUSP6. This is another point that, if addressed experimentally, could add value to the manuscript.

The ‘transient elevation’ in p-ERK with siRNA against DUSP6 may be a technical limitation of this assay as previously described. The time course studies in tetO lines and with BCI will help establish this. We will try and rescue the effects of siRNA against DUSP6 and BCI in H1975 cells—a cell line that is tolerant to ERK inhibition and provides an experimental system to ‘dial’ back appropriate p-ERK levels. A constitutively active ERK mutant is likely to be lethal in the presence of tet-induced mutant KRAS; for instance, ERK mutants have been documented to have a lethal effect in melanoma cells (Goetz et al., 2014). We will focus our attention on H1975 cells to rescue the effects of DUSP6 siRNA and BCI w/ MEK or ERK inhibition.

As mentioned above, we have used shRNA to inhibit ERK1 or ERK2 in PC9 cells and inhibition of ERK1 or ERK2 limited the toxic effects of knocking down DUSP6 (Figure 3—figure supplement 1G,H,I). In addition, we observed similar reductions in the toxic effects of BCI when H358 cells were co-treated with an ERK inhibitor (Figure 4—figure supplement 1E). These experiments further reinforce the conclusion that inhibition of DUSP6 (genetically or pharmacologically) decreases viability of EGFR or KRAS mutant cells – at least partially – through ERK induction.

h). If indeed it is true that a transient induction in pERK leads to cytotoxicity several days later, then does EGF stimulation (which also causes a transient induction in pERK) have the same effect in EGFR WT/KRAS MT cells?

As previously mentioned, the effects of transient vs. prolonged p-ERK will be addressed by studying the time course of response to BCI in cells and to mutant KRAS induced by dox. Transient treatment of cells with EGF is not expected to cause cytotoxicity based on our earlier experiments. In fact, transient administration of EGF to HCC95 cells failed to shift the IC50 for BCI; only prolonged exposure to EGF did that (Figure 5A). We will consider inclusion of this information during revision of the manuscript.

We have provided time course experiments of dox-mediated induction in TetO-KRAS cells (Figure 1—figure supplement 1B) and of BCI treatment in H358 cells (Figure 4—figure supplement 1D). Both approaches caused an initial increase in P-ERK levels coupled with later induction of cleaved-PARP and subsequent decrease in P-ERK (Figure 4—figure supplement 1D).

i) All three reviewers noted that HCC95 (RRID:CVCL_5137) is a squamous cell carcinoma line. This not comparable to lung adenocarcinoma because this tumor type has different pathway dependencies than LUAD (TCGA, 2012). A wild type LUAD line should be tested instead.

HCC95 cells were used because they are a cancer cell line (lung) that does not have mutations in EGFR or other examined components of the RAS pathway. We will state in the text that this is a squamous lung cancer cell line. Based on our data, cells with a mutation in the RAS pathway are likely be vulnerable to DUSP6 inhibition, regardless of cell lineage. Cells without these mutations (like HCC95) illustrate cancers (of any origin) that we would predict to be un-responsive to DUSP6 inhibition.

We have noted the nature of HCC95 cells in the text (subsection “Knockdown of DUSP6 elevates P-ERK and reduces viability of LUAD cells with either KRAS or EGFR oncogenic mutation”).

j) Figure 3C: Same comments as 3A.

Addressed above.

k) The investigators might consider inactivating DUSP6 with CRISPR-Cas9 in these cell lines to show genetic dependence.

We have now created PC9 and H358 cells in which *DUSP6* has been deleted or damaged using CRISPR-Cas9. We anticipate that the cells selected for this loss may have developed other mechanisms to maintain p-ERK or may have other pathways activated to bypass the need for p-ERK to mediate survival. Thus, they may or may not be uniquely vulnerable to inhibition of MEK or ERK. We will analyze the recently identified mutant cells for p-ERK levels, growth rates, and sensitivities to BCI and to inhibition of MEK and ERK.

We created H358 cells deficient in DUSP6 (Figure 4—figure supplement 1J,K). These cells were equally sensitive to BCI as cells that were targeted with a control (lacZ) sgRNA. We suspect that DUSP1 may be controlling p-ERK levels in the absence of DUSP6, and BCI has known specificity towards DUSP1 in addition to DUSP6. Additionally, these cell lines were derived from clones, so it is possible that new mutations or pathway re-wirings have taken place and that they continue to control p-ERK.

4) Figure 4 a) Figure 4A: 11 lung cancer cell lines were treated with BCI and viable cells were measured after 72-hour treatment. Most of these cell lines are fast growing cells with doubling time around 20~30 hours. Reduction of viable cells by 72-hours does not necessarily mean cell killing. TUNEL or caspase-3/PARP western blot needs to be performed to detect the levels of apoptosis. Senescence and cell cycle arrest should be examined too.

We agree that the reduction in numbers of viable cells does not reveal the mechanism by which the numbers were reduced, and we have been careful to avoid any suggestion that it does (e.g. by labeling our charts “number of viable cells”). As explained earlier, our focus has been on the generation of the toxic signal, not the response to it. Nevertheless, in response to the reviewers’ concerns on this point, we will examine cells for apoptosis by measuring PARP cleavage.

We have assessed cleaved-PARP after BCI treatment in a panel of seven sensitive and insensitive cell lines and in a time course experiment in H358s (Figure 4—figure supplement 1). Only sensitive cell lines induced cleaved PARP after treatment.

b) Figure 4C and D: p-ERK levels at a single time point was measured in multiple cell lines (30 min). In order to explain the loss of viable cells at 72hours in Figure 4A, p-ERK should be measured at serial time points such as 1, 6, 12, 24, 48, 72 hours.

We will provide a time course of our measurements of p-ERK and DUSP6 in H1975 and H358 cells during treatment with BCI.

We provide a time course for H358 cells treated with BCI at the time points suggested by the reviewer in Figure 4—figure supplement 1.

c) The data on BCI are very interesting but it's not clear how have the authors established that BCI is selective for DUSP? Sensitive and insensitive cell lines all have IC50s in the 1-5μM range (12/13 cells lines tested). What is the IC50 of the inhibitor in DUSP6 KD or KO cells (or in DUSP1/6 double KD/KO cells)? How do the authors know that the antiproliferative effect of BCI is due to its ability to induce pERK? These questions need to be addressed experimentally. The authors should also attempt to show that the inhibitor has an effect in vivo, given their claim that such an approach could be therapeutically beneficial in patients.

A potential target of BCI is DUSP1, but we have data showing that siRNA against DUSP1 is not lethal in H1975 cells while siRNA to DUSP6 is. We will now also try to rescue the effects of BCI in H1975 cells by inhibiting ERK, using this cell line for the reasons described above (3b,g). In addition to PC9 CRISPR-Cas9 DUSP6 knockout cells described above (Figure 3, comment k), we have generated PC9 cells expressing wild type and catalytically inactive DUSP6. We will use these cell models to determine if there is a shift in the BCI IC50 with manipulation of DUSP6 to further evaluate its role as the biological target of BCI.

Importantly, we have now completed a genome-wide CRISPR screen in H460 cells (an KRAS mutant cell line sensitive to BCI), looking for loss of function mutations that confer resistance to BCI. The most highly enriched guide RNA in this screen was specific for KRAS, suggesting that BCI is ‘on-target’ with respect to its proposed role in causing excessive activation of the RAS pathway. We will confirm these findings by measuring BCI sensitivity in H460 cells treated with siRNA against KRAS and expect to include a version of the results in the revised paper to further support our interpretation of our findings with BCI.

We have included data showing that DUSP1 knockdown, as opposed to DUSP6 knockdown, is not lethal in H1975 cells (Figure 4—figure supplement 1A,B). Further, we have demonstrated that co-treatment with an ERK inhibitor decreases the toxic effects of BCI in H358 cells (Figure 4—figure supplement 1E,F). Lastly, we added the genome-wide CRISPR screen data in H460 cells showing that sgRNAs for KRAS are enriched upon BCI treatment, which we have subsequently confirmed using individual sgRNAs and BCI dose response experiments. Together, these results confirm that BCI works mainly through DUSP6 in the context described and mediates its toxic effects through ERK.

d) Figure 4E and F: Four cell lines were treated with 125nM trametinib for 72 hours. 125nM is a very high dose for trametinib-sensitive cell lines such as H358 (reported IC50 ~50nM). Apoptosis levels should be measured to document any cell death.

The purpose of these experiments was to address whether there was a correlation between the sensitivity of cells to ERK inhibition *and* their sensitivity to ERK hyperactivation. This correlation appears to hold (PC9 and H358 vs. HCC95 and H1648).

5) Figure 5.See comments for Figure 3B. HCC95 is a SQCC cell line that is quite different from the LUAD cells used in the rest of this manuscript.a) In figures where the authors make a comparison of protein levels under different conditions the uncut blot with all the lanes on the same blot must be shown. It is not ideal to make a comparison between levels of proteins on two different strips of blots. For example, in Figure 5B, authors claim that EGF increase levels of pEGFR and pERK in HCC95 cells when the pEGFR and pERK levels with and without EGF are on two different strips of blots. This is important to confirm that BCI treatment increases pERK levels in EGFR treated cells. Quantifications should be shown in addition.

Data in the western blots are normalized to total ERK and actin to make the values comparable. This will be explicitly stated in methods. Additionally, samples grown with and without EGF will be run in the same gel for the highest dose of BCI to provide a visual comparison.

We have included a western blot using extracts of cells treated and not treated with EGF, run on the same gel in Figure 5—figure supplement 1.

Overall comment:The authors should soften earlier claims that EGFR-mutations and KRAS-mutations are synthetically lethal in the adenocarcinoma subtype of NSCLC (Unni et al., 2015). Recent genetic studies of EGFR-mutant lung cancer (Blakely et al., Nature Genetics 2017) have shown that 2.5%~4.7% EGFR-mutant lung cancer also harbor KRAS copy number gain or activating mutations. Tumor genomic analyses have indicated that bona fide driver mutations causing lung adenocarcinoma are not as mutually exclusive as previously thought (e.g. PMIDs: 25301630, 28498782, 28445112, 29106415, and other recent publications), particularly in metastatic lung adenocarcinoma instead of early-stage disease and/or during the evolution of treatment resistance in metastatic disease. Further, preclinical studies have shown acquired KRAS gain/activation in response to EGFRi in EGFR-mutant NSCLC, indicating that this can be a mechanism of drug resistance (Politi et al., 2000; Eberlein et al., 2015). The authors should acknowledge these recent works and temper their claim of absolute mutually exclusivity in this disease, at least under these more advanced-stage disease contexts and in the light of the emerging literature.

As noted earlier, we will provide a more thorough discussion of these points in the manuscript.

The additional discussion of these issues has been added to the text (Discussion section). We hope that with these changes and additions to the manuscript, the revised version will suitable for re-submission and consideration for publication at *eLife*.

[Editors' note: further revisions were requested prior to acceptance, as described below.]

Thank you for resubmitting your work entitled "Hyperactivation of ERK by mutation-driven RAS signaling or by inhibition of DUSP6 is toxic to lung adenocarcinoma cells" for further consideration at eLife. First, let me apologize profusely for the very long time this has taken. The problem is basically that the reviewers are at an impasse and could not decide on a course of action.Briefly, this is a follow up to a previous paper, that reported anti-proliferative effects of activation of Ras and EGFR in the same cancer cells. This new paper has provides evidence that the anti-proliferative effect of over-expressed, active Ras is due to ERK hyperactivation, and that DUSP6 is critical to prevent adverse effects of active ERK in lung cancer cells. In the revision, the authors have added results using ERK shRNA, ERKi, and a CRISPR screen that together provide convincing evidence that ERK mediates the antiproliferative effect of oncogene overexpression. The finding that sgKRAS are enriched in the BCI screen also indirectly supports that conclusion. The authors have also ruled out that the PI3K pathway is not involved by adding PI3K inhibitor experiments. These additions greatly strengthen the conclusion that hyperactivation of ERK is detrimental. However, the evidence that DUSP6 plays a key role is not convincing.The DUSP6 knockdown was done with a single siRNA with no rescue experiment. (Other siRNAs did not effectively inhibit DUSP6 expression). Attempts to recapitulate the result with a DUSP6 CRISPR knockout were unsuccessful. Therefore, the conclusion that DUSP6 is necessary relies in large part on the specificity of the chemical BCI. The DUSP6 KO cells have the same IC50 to BCI, while DUSP1 siRNA did not affect proliferation in their system. Based on these results, the authors cannot claim that the BCI effect is DUSP6 specific nor that the BCI effect in DUSP6KO cells is driven by DUSP1 inhibition. It probably isn't, and by inference, the observed phenotype is not dependent on DUSP6 alone but on other ERK specific DUSPs as well.The reviewers disagreed whether these weaknesses undermined the impact to the point where the paper was unsuitable for publication, or whether lengthy additional experiments would be needed. The eLife approach is not to ask for multiple rounds of revision. In this spirit, I suggest two possibilities:Either- Provide convincing evidence that validates DUSP6 as the key enzyme that downregulates ERK in lung cancer cells (e.g. try more siRNAs to find another that gives strong knockdown, and/or rescue DUSP6 siRNA with DUSP6 from mouse, or with silent mutations in the siRNA target sequence).Or,- Modify the title and abstract to allow for the possibility that other DUSPs are involved and be more open about the shortcomings of the results.

1) To test additional DUSP6 siRNAs for their effects on protein abundance and cell fitness, we obtained a DUSP6-specific siRNA from Qiagen (the previous siRNAs were prepared by Dharmacon). In contrast to the DUSP6-8 siRNA that targeted a sequence in the 3’ domain of DUSP6 mRNA, the new species (called DUSP6-Qiagen in Figure 3B,C) targeted a sequence in the 5’ coding region of DUSP6 mRNA and reduced levels of DUSP6 protein in PC9 cells to levels similar to those achieved with one of the previously tested Dharmacon siRNAs (DUSP6-8 in Figure 3B and Figure 3—figure supplement 1A) and with the pool of four Dharmacon siRNAs (DUSP6-pool in Figure 3B and Figure 3—figure supplement 1A). Furthermore, DUSP6-Qiagen reduced the number of viable PC9 cells to a level similar to that observed with the pooled Dharmacon siRNAs (Figure 3C). We have described the effects of this second effective inhibitory RNA in the text and conclude that it strengthens the case for a central role of DUSP6 in regulation of ERK activity in RTK-RAS-driven LUAD. We also point out that this conclusion is supported by the correlation between the effects of one Dharmacon siRNA (DUSP6-8) on both DUSP6 protein levels and cell fitness (Figure 3—figure supplement 1A,B).

2) We also attempted, unsuccessfully, to rescue the effects of DUSP6 siRNA by generating a plasmid encoding *DUSP6* mRNA with several synonymous mutations in the coding sequence to render the mRNA target sequence resistant to the siRNA without changing the protein sequence. For a variety of technical reasons related to transfection procedures, we have not been able to perform these experiments in a reproducible manner. We are convinced that the work required to carry out a satisfying rescue experiment would take an unreasonable amount of time and inappropriately delay publication, when we have provided the requested data with a second effective siRNA.

3) Despite our positive findings with the Qiagen siRNA, we recognize that our conclusions about the role of DUSP6 in regulation of the activity of ERK kinases should be cautious. (DUSP6 may not be the only important regulator and we cannot fully exclude some off-target effects of our siRNAs.) We have therefore removed specific mention of DUSP6 in the title of the manuscript, and we have modulated the description of the results in the abstract, along the lines suggested in your letter.

One other relevant item: two recent papers confirm the significance of the level of ERK kinase activity in another cancer type, melanoma, and address the possible role of DUSP6. Leung et al., over-express *ERK2* in melanoma cell lines and show that high levels of ERK2 protein are toxic specifically in lines that carry BRAF V600E. Wittig-Blaich et al., use a complex screening method to identify genes that produce a synthetic lethality when disrupted in melanoma cell lines carrying the BRAF V600E mutation; one of the five implicated genes is *DUSP6*, allowing the authors to draw conclusions similar to our own. We mention and cite these papers (Leung et al., 2018 and Wittig-Blaich et al., 2017) in the Discussion section.

In addition to the changes that address your main concerns, we have found a few places in the text that lacked clarity upon careful re-reading of our previously submitted revision.